# LLMs Reading the Rhythms of Daily Life: Aligned Understanding for Behavior Prediction and Generation

## Abstract

Human daily behavior unfolds as complex sequences shaped by intentions, preferences, and context. Effectively modeling these behaviors is crucial for intelligent systems such as personal assistants and recommendation engines. While recent advances in deep learning and behavior pre-training have improved behavior prediction, key challenges remain—particularly in handling long-tail behaviors, enhancing interpretability, and supporting multiple tasks within a unified framework. Large language models (LLMs) offer a promising direction due to their semantic richness, strong interpretability, and generative capabilities. However, the structural and modal differences between behavioral data and natural language limit the direct applicability of LLMs. To address this gap, we propose Behavior Understanding Alignment (BUA), a novel framework that integrates LLMs into human behavior modeling through a structured curriculum learning process. BUA employs sequence embeddings from pretrained behavior models as alignment anchors and guides the LLM through a three-stage curriculum, while a multi-round dialogue setting introduces prediction and generation capabilities. Experiments on two real-world datasets demonstrate that BUA significantly outperforms existing methods in both tasks, highlighting its effectiveness and flexibility in applying LLMs to complex human behavior modeling. The code is available at https://anonymous.4open.science/r/dasjijio-21B2/

## 1 Introduction

Human daily life unfolds as a sequence of behaviors—ranging from habitual routines to spontaneous actions—each reflecting underlying intentions, preferences, and contextual factors. Accurately modeling and understanding these human daily behaviors is fundamental to a wide range of intelligent systems, including personalized assistants, recommender engines, and context-aware services (Chung & Lee, 2018; Tulshan & Dhage, 2019; Savcisens et al., 2023). Traditional approaches (Zhu et al., 2017; Chen et al., 2018; Yuan et al., 2023), particularly those based on deep learning, have primarily focused on behavior prediction: learning to predict the next event based on historical sequences (Kang & McAuley, 2018; Sun et al., 2019). Recently, with the increasing availability of large-scale behavioral datasets and inspired by the success of pre-training paradigms in natural language processing (NLP) (Radford et al., 2019; Dubey et al., 2024), behavior pre-training has emerged as a promising technique. These methods (Gong et al., 2024; Savcisens et al., 2024; Zhai et al., 2024) exploit vast human daily behavioral corpora to capture intricate temporal dependencies and latent patterns, leading to significant improvements in predictive accuracy.

Despite these advances, existing human daily behavior modeling approaches suffer from several **fundamental limitations**. First, they struggle to model **long-tail behaviors**—actions that occur infrequently or are newly emerging—due to inherent data sparsity issues (Hu et al., 2025; Kim et al., 2024). Second, their "black-box" nature offers limited insight into the decision-making process, creating a gap between the model's predictions and **human-interpretable reasoning** (Lei et al., 2024). Third, most models are designed for a single task, focusing on either prediction or generation, and lack the flexibility to handle both within a **unified framework**.

Recent developments in Large Language Models (LLMs) offer a powerful new direction for addressing these challenges. LLMs provide several distinct advantages: (1) Their rich semantic representations, learned from vast textual corpora, can enhance the modeling of long-tail behaviors by providing crucial contextual understanding (Liu et al., 2024; Sheng et al., 2024). (2) Trained on extensive human-generated text, LLMs can process and articulate behavioral patterns in a textual format that aligns more closely with human cognition, thereby enhancing model interpretability. (3) Their **inherent generative capabilities** support multitask learning through natural language, enabling both behavior prediction and generation within a single, unified model. Overall, integrating LLMs presents a clear opportunity to overcome the core limitations of traditional behavior modeling.

However, a critical modality gap exists: human behavioral data, typically represented as sequences of IDs or embeddings, is structurally and semantically different from the natural language data LLMs are trained on. Consequently, LLMs cannot directly interpret the feature representations or outputs of conventional behavior modeling pipelines. To bridge this gap, we propose the **Behavior Understanding Alignment (BUA)** framework. BUA is a novel approach that unlocks the potential of LLMs for both behavior prediction and generation by first teaching the LLM to *understand* human behavior sequences through a structured alignment process.

Our framework leverages sequence embeddings from a pretrained behavior model as alignment anchors and guides the LLM through a structured three-stage curriculum. This curriculum is designed to progressively bridge the modality gap, beginning with basic sequence comprehension and advancing to more complex predictive and generative reasoning. Furthermore, we introduce a multi-round dialogue setting that establishes a coherent reasoning chain. This process compels the LLM to first generate an explicit textual summary of its understanding of a given behavior sequence. This summary then acts as a contextual foundation, or a cognitive "scaffold", from which the model subsequently performs prediction and generation tasks, significantly enhancing the performance of both.

The contributions of this work are summarized as follows:

- We are the first to propose training an LLM to explicitly *understand* human daily behavior sequences—by aligning behavioral and language modalities—as a foundational step for improving downstream prediction and generation tasks.

- We introduce the Behavioral Understanding Alignment (BUA) framework, which uniquely combines a three-stage curriculum learning pipeline with a multi-round dialogue mechanism to synergistically enhance the model's capabilities in understanding, predicting, and generating human behaviors.

- Experimental results on two real-world datasets demonstrate that BUA achieves state-of-the-art performance in both prediction and generation tasks. Comprehensive ablation studies further validate the critical role of our structured curriculum and dialogue-based reasoning process in achieving these results.

## 2 RELATED

### 2.1 BEHAVIOR MODELING

Modeling daily human behavior hinges on capturing core patterns in user behavior sequences, typically through two tasks: behavior prediction and behavior generation. Early behavior prediction models, such as TRNN (Zhu et al., 2017), utilized time-difference-aware embeddings to enhance temporal modeling. As datasets expanded, transformer-based pretraining methods like BehaveGPT (Gong et al., 2025) and Life2Vec (Savcisens et al., 2024) became prevalent, significantly improving predictive accuracy. However, these methods often struggle with long-tail behaviors due to limited sample diversity. For behavior generation, early rule-based and agent-based models (Kim et al., 2019; Pfoser & Wenk) relied on hand-crafted logic, limiting their ability to capture real-world complexity. SAND (Yuan et al., 2023) advanced this by using neural stochastic differential equations, enabling more realistic dynamics without fixed rules, though its static generation parameters limit adaptability. More recently, D2A (Wang et al., 2024) trained an LLM as a cognitively inspired agent guided by a dynamic value system, enhancing behavioral diversity and flexibility. However, it

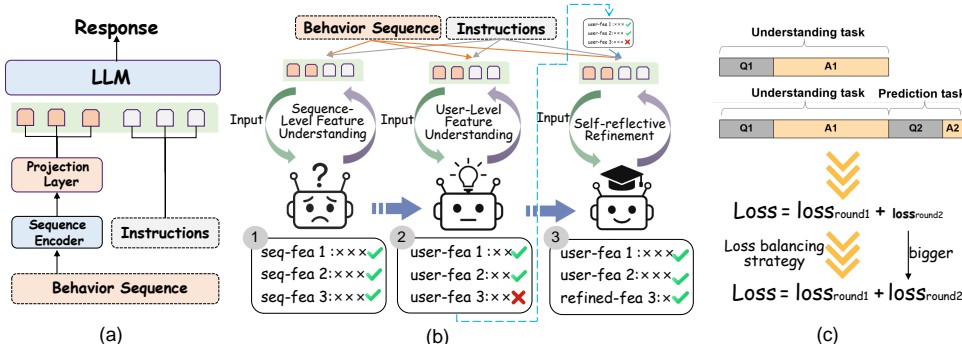

Figure 1: The framework of Behavior Understanding Alignment (BUA). (a) the modality conversion process using sequence embedding. (b) the structured curriculum learning process: seq-fea, user-fea, and refined-fea represent features learned in Stage 1 (Sequence-Level), Stage 2 (User-Level), and Stage 3 (Self-Reflection). The ✓ and × marks indicate the correctness of the learned features. (c) Understanding-enhanced prediction and generation via multi-round dialogue.

underutilizes the LLM's potential for sequence generation based on a deep, multimodal understanding of behavioral context.

## 2.2 ALIGNMENT IN RECOMMENDATION

Our behavior prediction task is defined as predicting the next behavior based on the user's most recent $L$ behavioral events, which is similar to sequential recommendation. The two differ in focus: behavior prediction emphasizes recurring daily behaviors, whereas sequential recommendation often targets novel items. To the best of our knowledge, no prior work has focused on explicitly aligning LLMs with the underlying semantic representations of entire behavior sequences, so we draw upon related research in sequential recommendation. Existing work that incorporates large language models (LLMs) into sequential recommendation can be broadly categorized into two directions: (1) LLMs as standalone recommendation systems (Tan et al., 2024; Kim et al., 2024; Liao et al., 2024; Zhang et al., 2025), and (2) LLMs as enhancers of traditional systems (Ren et al., 2024; Liu et al., 2024; Hu et al., 2025; Wang et al., 2024).

For the first category, these works treat recommendation as a text generation problem but often bypass a deeper alignment with the rich, latent representations of user behavior learned by specialized encoders. Their item-embedding alignment-based approach thus fails to fully leverage collaborative knowledge and complex sequential patterns. The latter line of work offers superior inference efficiency compared to fully LLM-based systems. However, by using the LLM as a supplementary component rather than the core reasoning engine, they inherit the limitations of traditional deep learning models, including poor generalization to new items and tasks, and limited interpretability due to their black-box nature (Lei et al., 2024). In contrast, our approach trains the LLM to first *understand* behavior sequence embeddings, positioning it as the central agent for both prediction and generation.

## 3 METHOD

### 3.1 PROBLEM FORMULATION

Let $x_i$ denote a basic behavioral event, represented as a four-tuple $(d_i, t_i, l_i, b_i)$, where $d_i$ is the Day of week index, $t_i$ the Timestamp ID, $l_i$ the Location ID, and $b_i$ the Behavior Type ID. The behavior type captures high-level daily activities—such as exercising or gaming—rather than fine-grained actions. We consider two related tasks under this representation:

**Behavior Prediction and Generation.** Given a user's recent behavior sequence $X_{\text{seq}} = \{x_1, x_2, \ldots, x_{L_1}\}$, the model is tasked with either (1) predicting the next behavior $b \in \mathcal{B}$, or (2) generating a future sequence $Y_{\text{seq}} = \{y_1, y_2, \ldots, y_{L_2}\}$, where each $y_i$ is a four-tuple $(d_i, t_i, l_i, b_i)$.

## 3.2 OVERVIEW

The framework of our method Behavior Understanding Alignment (BUA) is shown in Figure 1. We propose a three-stage structured curriculum to progressively enhance behavior understanding: (1) Sequence-Level Understanding, (2) User-Level Feature Modeling, and (3) Self-Reflective Refinement, with sequence embedding as the alignment anchor. Additionally, in the second stage, we incorporate a multi-round dialogue setup that integrates prediction and generation tasks.

## 3.3 SEQUENCE-EMBEDDING-BASED ALIGNMENT

Given a user behavior sequence $X_{\text{seq}}$, we first encode it using BehaveGPT (Gong et al., 2025), a pretrained model on large-scale behavioral data, serving as the behavior encoder $g_\phi$. The penultimate hidden state is then projected into a unified representation space via a lightweight two-layer MLP, producing the behavior sequence embedding $H_{\text{seq}}$:

$$H_{\text{seq}} = \text{MLP}(g_\phi(X_{\text{seq}})) \tag{1}$$

This embedding is then concatenated with the encoded text instruction $X_{\text{Ins}}$ and input into the LLM, which generates a textual response $y$, as shown in Figure 1(a). All our fine-tuning tasks are optimized using the following objective, which maximizes the likelihood of the target output given the multimodal input:

$$\mathcal{L} = -\frac{1}{N} \sum_{i=1}^{N} \log P_\theta \left( y_t \mid y_{<t}, X_{\text{Ins}}, X_{\text{seq}} \right) \tag{2}$$

where $N$ denotes the length of the response $y$.

## 3.4 CURRICULUMN FOR BEHAVIOR UNDERSTANDING ALIGNMENT

We propose a three-stage curriculum for user behavior understanding tasks, structured to progress from simple to complex. The first stage targets sequence-level understanding, the second emphasizes user-level feature modeling, and the third incorporates self-reflective refinement to further enhance behavioral representations. The full task structure is detailed in Appendix B.

### 3.4.1 STAGE 1: SEQUENCE-LEVEL FEATURE UNDERSTANDING

In this initial stage, the model learns to interpret behavioral sequence embeddings from language modalities, building a foundation for deeper behavioral understanding in later stages. Based on empirical insights, we introduce three basic tasks:

- **Historical Sequence Reconstruction**: The model reconstructs the original behavioral sequence in natural language, capturing key temporal, spatial, and behavioral transitions. This entry-level task establishes the groundwork for multimodal understanding.

- **Current Scene Summary**: The model summarizes the user's recent context over the past two hours (e.g., morning commute), requiring it to extract and generalize key patterns, advancing its sequence-level understanding.

- **Future Scene Inference**: The model predicts the user's likely context in the next two hours (e.g., evening commute or pre-bedtime leisure), demonstrating its ability to analyze sequence dynamics and temporal trends.

In practice, we also introduce simple user-level inference tasks at this stage, such as home/workplace location identification and user hobby inference, which provide a natural transition to the more complex user-level understanding required in Stage 2. Additionally, following common acceleration training techniques for large multimodal language models, we freeze both the sequence encoder and the LLM during this stage, allowing only the parameters of the projection layer (MLP) to remain trainable. While this approach constrains model adaptability compared to full-parameter fine-tuning, it represents a strategic trade-off that significantly enhances training efficiency without compromising the effectiveness of feature alignment.

### 3.4.2 STAGE 2: USER-LEVEL FEATURE UNDERSTANDING

In this phase, the focus shifts to capturing deeper user-level features embedded within behavioral sequences. It is not sufficient for the model to recognize surface-level changes in time, location, or activity; instead, it must abstract the underlying user features that drive these patterns. Such features are critical for understanding behavior trends and informing prediction and generation tasks. Inspired by how humans infer user features from behavioral sequences, we design the following tasks to guide user-level feature learning:

- **User Key Behavior Identification**: The model identifies semantically rich behaviors that are frequent or mark transitions between daily phases (e.g., taking the subway after work indicates a shift from work to evening leisure). These behaviors are critical for inferring user intent development.
- **User Behavior Pattern Discovery**: The model detects recurring behavioral subsequences and consistent temporal-spatial patterns (e.g., watching TikTok during commutes). These patterns reveal deeper user preferences and routines.
- **User Feature Summarization**: The model abstracts high-level user features (e.g., This user prefers light entertainment during their evening commute), which provides a higher-level, more essential understanding of user behavior features.

These tasks are intentionally sequenced from simple to complex, forming a structured learning path. To support effective user-level understanding, this progression is enforced during training: the model performs User Key Behavior Identification first, followed by Behavior Pattern Discovery, and finally User Feature Summarization. Additionally, during training at this stage, we freeze the sequence model parameters while allowing the parameters of the projection layer and LLM to be adjustable.

### 3.4.3 STAGE 3: SELF-REFLECTIVE REFINEMENT

In this phase, we introduce Self-Reflective Refinement to enhance the model's understanding of user features. After the first two phases, the model's performance on the User Feature Summarization task remained suboptimal. We evaluated the generated summaries and manually inspected low-scoring responses, finding that there are some recurring issues, such as unclear relationships between behavioral features. However, as these issues did not stem from fundamental misunderstanding, the model might have developed a comprehensive understanding of user features during the Sequence-Level Understanding and User-Level Feature Extraction stages. Instead, the model only requires more "thinking" to generate reasonable and accurate user features. To address this, we propose a self-reflective iteration strategy that empowers the model to identify and correct its own shortcomings. Specifically, we summarize the recurring issues, and designed targeted correction criteria, guiding the model to review and revise its earlier outputs based on clear feedback. The task is defined as follows:

- **Self-Reflective Refinement**: The model reviews low-quality user feature summaries, identifies key issues, and refines them using its understanding of sequence embeddings, producing more accurate and coherent summaries.

Based on the foundations established in the first two stages, this strategy leverages the model's reasoning capabilities for iterative improvement, resulting in more robust and accurate user feature representations. During this phase, we freeze the sequence encoder and projection layer while keeping LLM parameters trainable.

### 3.5 UNDERSTANDING-ENHANCED PREDICTION AND GENERATION VIA MULTI-ROUND DIALOGUE

To develop behavior prediction and generation capabilities, we introduce a multi-round dialogue framework in the second stage of the behavior understanding pipeline. This approach enables the model to simultaneously refine prediction and generation skills while deepening its understanding of user behavior features. In this setup, the model starts with the Key Semantic Behavior Recognition task in the first round, progressively completes all user-level behavior understanding tasks, and concludes with the corresponding prediction or generation tasks in the final round. By leveraging

intermediate understanding and analysis from earlier rounds, the model enhances the accuracy and effectiveness of downstream tasks. The optimization loss for this multi-round dialogue setting is defined as:

$$Loss = \text{mean}\left(\sum_{i=1}^{N}\sum_{t=1}^{T_i}\log P_\theta\left(y_t^{(i)}\big|y_{<t}^{(i)}, X_{\text{Ins}}^{(i)}, Y^{(i)}, X_{\text{seq}}\right)\right) \tag{3}$$

where $N$ is the total number of dialogue rounds, $T_i$ is the number of tokens in the answer for the $i$-th round, $\theta$ denotes the LLM parameters, $y_t^{(i)}$ is the $t$-th token of the $i$-th round's output, $X_{\text{Ins}}^{(i)}$ is the input Instruction, and $Y^{(i)}$ represents the corresponding answer for the $i$-th round.

However, multi-round dialogues risk imbalanced training across different rounds. Rewriting the loss from the token level to the round level, we get

$$Loss_{\text{multi-turn}} = -\sum_{i=1}^{N}\frac{T_i}{\sum_{i=1}^{N}T_i}loss_{\text{i}} \tag{4}$$

where $loss_i$ denotes the average loss for the $i$-th round. This means that rounds with longer answers dominate the overall loss, while those with shorter outputs receive less attention at the round level. This is problematic in practice, as understanding tasks typically involve long outputs (often exceeding 100 tokens), whereas prediction tasks only output the predicted behavior type (often fewer than 5 tokens). As a result, the model struggles to effectively learn shorter prediction tasks.

To address this issue, we introduce a simple yet effective loss balancing strategy that ensures equal attention across rounds. Specifically, we apply a weight $W_i = \frac{\sum_{i=1}^{N}T_i}{NT_i}$ that is inversely proportional to the length of the answer in each round, encouraging balanced learning across all rounds. The final loss function becomes:

$$Loss_{\text{weighted}} = -\frac{1}{N}\sum_{i=1}^{N}\log P_\theta\left(y_t^{(i)}\,\Big|\,y_{<t}^{(i)},\,X_{\text{Ins}}^{(i)},\,Y^{(i)},\,X_{\text{seq}}\right) \tag{5}$$

This balanced loss formula significantly enhances the performance of the model on the behavior prediction task without significantly reducing the effectiveness on the understanding task.

## 4 EXPERIMENT

### 4.1 EXPERIMENTAL SETTINGS

**Datasets**. We evaluated our model on two real-world user behavior datasets: **Behavior dataset**: This dataset is derived from the user's mobile phone logs. After desensitization, it includes 37 daily behaviors that cover a wide range of life scenarios, including activities related to learning, work, entertainment, leisure, and more. **Tencent Dataset (Shao et al., 2024)**: This dataset is derived from the user's social network and the user's movement trajectory. It includes 14 human behavior intentions, such as eating, going home, working, etc.

For both datasets, we split the users in a ratio of 8:1:1 to create training, validation, and test datasets. For more detailed information about the datasets and their splits, please refer to the Appendix C.

**Evaluation Metrics**. For **behavior prediction task**, we adopt commonly used metrics, weighted precision ($Prec_w$) and weighted recall ($Rec_w$)(equivalent to HR@1), to evaluate the overall prediction performance of the model. Additionally, user data is often unevenly distributed and exhibits a clear long-tail distribution in practice (Kim et al., 2024). Following relevant work (Liu et al., 2019; Shi et al., 2024), we construct a long-tail intent test set and adopt global accuracy(denoted as $Overall$), high-frequency behavior accuracy(denoted as $Head$), medium-frequency behavior accuracy(denoted as $Medium$), and long-tail behavior accuracy (denoted as $Tail$) to fairly evaluate the model's performance across different behavior categories. For the calculation methods and additional details on all six metrics, please refer to the Appendix D. For **behavior generation task**, we adopt commonly used metrics $BLEU, TVD$, and $JSD$ to measure the time, location, and behavior

Table 1: Experiment results on next behavior prediction

| Category | Method | Honor Dataset | | | | | | Tecent Dataset | | | | | |
|---|---|---|---|---|---|---|---|---|---|---|---|---|---|
| | | $Rec_w$ | $Prec_w$ | $Overall$ | $Head$ | $Medium$ | $Tail$ | $Rec_w$ | $Prec_w$ | $Overall$ | $Head$ | $Medium$ | $Tail$ |
| Traditional | SASRec | 0.546 | 0.535 | 0.291 | 0.420 | 0.340 | 0.222 | 0.328 | 0.269 | 0.097 | 0.29 | 0.045 | 0.021 |
| | BehaveGPT | 0.567 | 0.551 | 0.206 | 0.442 | 0.354 | 0.027 | 0.509 | 0.426 | 0.113 | 0.537 | 0 | 0 |
| LLM-Enhanced | PITuning | 0.617 | 0.603 | 0.408 | 0.481 | 0.444 | 0.361 | 0.524 | 0.466 | 0.120 | 0.546 | 0.009 | 0.0 |
| | AlphaFuse | 0.578 | 0.575 | 0.242 | 0.457 | 0.380 | 0.075 | 0.507 | 0.435 | 0.118 | 0.547 | 0.001 | 0 |
| LLM-Based | Deepseek-V3 | 0.492 | 0.495 | 0.237 | 0.330 | 0.265 | 0.191 | 0.318 | 0.282 | 0.119 | 0.303 | 0.083 | 0.038 |
| | TALLRec | 0.617 | 0.607 | 0.398 | 0.452 | 0.434 | 0.355 | 0.561 | 0.543 | 0.134 | 0.513 | 0.044 | 0.019 |
| | A-LLMRec | 0.584 | 0.557 | 0.348 | 0.422 | 0.394 | 0.299 | 0.539 | 0.523 | 0.140 | 0.542 | 0.037 | 0.025 |
| | CoLLM | 0.618 | 0.596 | 0.408 | 0.453 | 0.448 | 0.363 | 0.560 | 0.543 | 0.152 | 0.530 | 0.067 | 0.034 |
| | LLaRA | 0.615 | 0.608 | 0.404 | 0.462 | 0.439 | 0.361 | 0.564 | 0.545 | 0.152 | 0.543 | 0.060 | 0.033 |
| | **BUA** | **0.644** | **0.642** | **0.471** | **0.538** | **0.489** | **0.446** | **0.600** | **0.574** | **0.207** | **0.62** | **0.114** | **0.041** |
| | Improv | 4.2% | 5.6% | 15.4% | 11.9% | 9.2% | 22.9% | 6.4% | 5.3% | 36.2% | 13.4% | 37.4% | 7.9% |

similarity between the generated sequence and the real sequence data. The calculation methods for the metrics are outlined in the Appendix E.

**Baselines**. For **behavior prediction task**, We selected representative algorithms from various categories to compare with our proposed algorithm. For traditional deep learning methods, we chose SASRec (Kang & McAuley, 2018), BehaveGPT (Gong et al., 2025). For pure LLM-based prediction methods, we selected DeepSeek-V3 (DeepSeek-AI, 2025) and TallRec (Bao et al., 2023). For methods that use modality fusion and LLM as recommendation systems (similar to ours), we selected A-LLMRec (Kim et al., 2024), CoLLM (Zhang et al., 2025), and LLaRa (Liao et al., 2024). For methods that employ modality fusion and LLM as recommendation system enhancers, we selected PI-Tuing (Gong et al., 2024) and AlphaFuse (Hu et al., 2025). For **behavior generation task**, We chose SAND (Yuan et al., 2023), a representative method based on deep learning, and D2A (Wang et al., 2024), which uses LLM for user behavior activity generation based on Maslow's Theory. For further details on the baselines, please refer to the Appendix F.

**Implementation Details**. The hardware used in this experiment consists of 8 NVIDIA A100 40G GPUs. We chose Qwen2.5-7B (Team, 2024) as the backbone model for the experiment. More details about the implementation are in the Appendix G.

## 4.2 BEHAVIOR PREDICTION EXPERIMENT

To validate the effectiveness of our method, we evaluated the model on two real-world datasets against the baselines. Our model consistently outperformed all baselines across all metrics, confirming its effectiveness, as shown in Figure 1. We further probe more conclusions by the following analysis.

**Overall Comparison**. The results show that our method(BUA) outperforms all baselines on both $Prec_w$ and $Rec_w$ under the real data distribution. Additionally, most LLM-based methods surpass traditional models like SASRec, underscoring the value of semantic information in behavior prediction. Notably, item embedding fusion approaches (e.g., LLARA, A-LLMRec) offer no clear advantage over the pure LLM method, TallRec, indicating that item embeddings alone are insufficient to fully leverage sequential knowledge.

**Different Categories of Behaviors Comparison**. The results show that our method achieves substantial gains across high-frequency, medium-frequency, and long-tail behaviors, with an overall average improvement of 25.8% over the best baseline on both datasets. While the LLM-based TallRec outperforms SASRec by over 50% on long-tail behaviors, it shows only a 10% gain on high-frequency ones on Behavior dataset, emphasizing the importance of semantic information for long-tail prediction. Although the item-embedding fusion method LLARA slightly outperforms the pure LLM TallRec, the margin is small compared to our approach—further confirming the advantage of our method.

Table 2: Experiment result on behavior sequence generation

| Method | Event | | | Timestamp | | | Location | | |
|---|---|---|---|---|---|---|---|---|---|
| | Bleu ↑ | TVD ↓ | JSD ↓ | Bleu ↑ | TVD ↓ | JSD ↓ | Bleu ↑ | TVD ↓ | JSD ↓ |
| BehaveGPT | 0.009 | 0.945 | 0.632 | — | — | — | — | — | — |
| SAND | 0.142 | 0.304 | 0.083 | 0.344 | 0.204 | 0.038 | — | — | — |
| D2A | 0.315 | 0.183 | 0.039 | 0.287 | 0.223 | 0.049 | 0.396 | 0.529 | 0.173 |
| Ours | 0.354 | 0.140 | 0.020 | 0.541 | 0.147 | 0.020 | 0.711 | 0.065 | 0.005 |

### 4.3 BEHAVIOR GENERATION EXPERIMENT

We evaluated our method on the Behavior dataset against all baselines. To ensure fair comparison with SAND, which outputs fixed-time behaviors, we generated one day of future behaviors. Our model consistently outperformed all baselines across all metrics, demonstrating strong robustness and effectiveness (see Figure 2). Note that − in table indicates the model lacks generation capability and is not applicable for evaluation. Additionally, we found that BehaveGPT, despite large-scale pretraining, performs poorly on the generation task even after fine-tuning, revealing limited capability. While SAND generates more accurate timestamps than D2A, it lags in behavior accuracy. These results underscore the importance of behavioral semantics and the difficulty LLMs face with temporal and numerical features. Our method addresses this by first understanding and summarizing behavioral patterns, leading to superior performance.

### 4.4 SYNTHETIC DATA FOR DOWNSTREAM PREDICTION TASK

To further assess the usability of the generated data, following (Yuan et al., 2023), we evaluated our model in a hybrid setting that augments real data with synthetic data. Using the standard SASRec model for next-behavior prediction (see Appendix H). As shown in Figure 2, our generated data consistently outperforms the strongest baseline, D2A, in both overall and average accuracy (Figure 2), significantly boosting

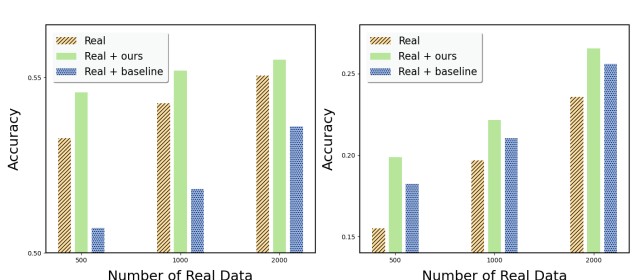

Figure 2: hybrid scenario

model performance. This confirms our model's ability to generate high-fidelity behavior sequences that capture underlying user patterns.

### 4.5 CROSS-MODEL ENHANCEMENT VIA BEHAVIORAL UNDERSTANDING TRANSFER

To further validate the effectiveness of our model's behavior understanding, we evaluate whether its extracted user features could enhance other models. These features were added to TallRec and LLARA, resulting in TallRec-cross and LLARA-cross. Specifically, BUA generated user summaries for 20,000 samples to

Table 3: Cross-Model enhancement via behavioral understanding transfer

| Method | Weighted Metrics | | Category Performance | | | |
|---|---|---|---|---|---|---|
| | $Rec_w$ | $Prec_w$ | $Overall$ | $Head$ | $Medium$ | $Tail$ |
| Tallrec | 0.609 | 0.587 | 0.384 | 0.460 | 0.440 | 0.319 |
| **Tallrec-cross** | **0.620** | **0.610** | **0.405** | **0.467** | **0.463** | **0.341** |
| Llara | 0.605 | 0.585 | 0.362 | 0.460 | 0.412 | 0.296 |
| **Llara-cross** | **0.610** | **0.595** | **0.403** | **0.468** | **0.466** | **0.335** |

supplement each model's input. As shown in Table 3, this consistently improved prediction performance, with notable gains on long-tail behaviors.

### 4.6 ABLATION STUDY

We conduct an ablation study on the Behavior dataset to evaluate the influence of different design components on overall performance. Specifically, we assess the model's performance under the

following conditions: (1) Removal of the first stage: Sequence-Level Feature Understanding (w/o stage1), (2) Removal of User-Level Feature Extracting in second stage (w/o stage2), and (3) Removal of the third stage: Self-reflective Refinement(w/o stage3)

**Behavior Prediction** To further analyze the contributions of each component to the Behavior Prediction task, we additionally evaluate: (4) Removal of loss balancing strategy of multi-turn dialogue in the second stage (w/o loss balance), (5) Use of item embedding instead of sequence embedding for modality alignment (item-emb). The results are shown in Table 4, with key findings as follows: All components contribute to overall performance, with the loss balancing strategy in the second-stage multi-turn dialogue having the most significant impact on prediction. Without it, the model favors the understandng task due to its longer output, reducing prediction effectiveness. Although the first and third stages do not directly target prediction, they improve behavior sequence understanding, indirectly enhancing prediction. Finally, replacing sequence embeddings with item embeddings for modality fusion leads to performance degradation, confirming the superiority of sequence-level representations.

**Behavior Generation** All design components contribute to the model's overall performance. Removing the user-level feature extraction in the second stage has the greatest impact, showing that explicitly generating user behavior features enhances understanding and guides future behavior generation. Additionally, the behavior understanding tasks in the

Table 4: Ablation study for behavior prediction task

| Method | Weighted Metrics | | Category Performance | | | |
|---|---|---|---|---|---|---|
| | $Rec_w$ | $Prec_w$ | $Overall$ | $Head$ | $Medium$ | $Tail$ |
| **Ours** | **0.644** | **0.642** | **0.471** | **0.538** | **0.489** | **0.446** |
| w/o stage1 | 0.613 | 0.607 | 0.370 | 0.485 | 0.452 | 0.286 |
| w/o stage2 | 0.587 | 0.577 | 0.456 | 0.400 | 0.159 | 0.291 |
| w/o stage3 | 0.592 | 0.586 | 0.334 | 0.420 | 0.357 | 0.300 |
| w/o loss balance | 0.560 | 0.552 | 0.285 | 0.385 | 0.364 | 0.197 |
| item-emb | 0.631 | 0.626 | 0.465 | 0.523 | 0.491 | 0.437 |

first and third stages improve the model's grasp of behavior sequences, as their removal weakens pattern understanding and indirectly reduces generation performance. Please refer to Appendix I for detailed results.

## 4.7 QUALITATIVE AND QUANTITATIVE ANALYSIS OF INTERPRETABILITY

To comprehensively evaluate the model's understanding and interpretability capabilities, we conducted both qualitative case studies and quantitative human evaluations.

**Qualitative Analysis: Evolution of User Profiling.**

We demonstrate the model's ability to capture semantic modalities through the evolution of User Feature Summarization across the three curriculum stages. For clarity, we present only a representative sub-feature, as full outputs are too lengthy.

> *Stage 1: The user frequently reads news throughout the day and is a news enthusiast.*
> *Stage 2: News Enthusiast: The user has a strong habit of consuming news, **often checking it multiple times in quick succession**. This suggests **a desire to stay informed** about current events.*
> *Stage 3: Information-Seeking Behavior: The user has a strong habit of **consuming news** and **checking the weather**, indicating **a desire to stay informed** about current events and environmental conditions. This behavior is **consistent throughout the week, with slight variations in timing**.*

The outputs show clear improvement across training stages. After the first stage, the model produces simple summaries (e.g., "the user likes to read news"). By the end of the second stage, it begins identifying behavioral patterns like "checking multiple times in rapid succession" and inferring intent. In the third stage, the model generates more comprehensive features, grouping behaviors such as "reading news" and "checking the weather" into higher-level categories like "information-seeking behavior" with richer detail. These results highlight the effectiveness of our three-stage training framework.

**Quantitative Analysis: Human Evaluation.** To more accurately assess the quality and interpretability of the behavioral features generated by our model, we conducted a systematic human evaluation following standard protocols (Lei et al., 2024). We randomly selected 120 test samples from the Honor dataset. For each sample, user features were generated by three sources: (1) Human

annotators, (2) BUA (Ours), and (3) the Base Model (Qwen2.5-7B without fine-tuning). A group of human judges evaluated these blinded samples on a 0–3 scale based on two dimensions:

- **Rationality**: The degree to which the features align with the user's historical behavior.

- **Interpretability**: The extent to which the features help explain the user's predicted future behavior.

Table 5: Average Scores (0–3 Scale)

| Type | Rationality | Interpretability |
|------|-------------|------------------|
| Human | 2.51 | 2.55 |
| BUA | 2.46 | 2.39 |
| No Tune | 1.90 | 1.83 |

Table 6: Best Score Probability

| Metric | Human | BUA | No Tune |
|--------|-------|-----|---------|
| Rationality | 39.8% | 37.9% | 22.3% |
| Interpretability | 45.3% | 33.3% | 21.4% |

The results are summarized in Table 5 and Table 6. While human-written features achieve the highest performance, BUA significantly outperforms the Base Model, improving the interpretability score from 1.83 to 2.39. Furthermore, regarding the "Best Score Probability", which measures how often a model's output was rated as the best among all candidates, BUA achieves competitive results comparable to human annotations (e.g., 37.9% vs. 39.8% in Rationality). These results quantitatively demonstrate that BUA acquires a deep, alignable understanding of user behavior that approaches human-level. For more detailed experimental setup and interpretability question settings, please refer to Appendix Q and Appendix R respectively.

## 5 CONCLUSION

This paper presents the Behavior Understanding Alignment (BUA) framework, which integrates large language models into human behavior modeling using structured curriculum learning. BUA addresses key limitations of traditional models—such as poor long-tail predictions and limited interpretability—by aligning behavior and language through sequence embeddings from pretrained models.

## ETHICS STATEMENT

We implemented robust measures to ensure ethical data handling throughout the research process, prioritizing privacy, security, and bias mitigation. To protect individual privacy, trajectory data is strictly anonymized and contains no personally identifiable information (PII). We took several steps to address privacy and ethical considerations:

- **Anonymization procedure**: Each user is assigned a random, anonymous ID, which is regenerated every three months to prevent long-term tracking.

- **Location data processing**: We do not collect precise location coordinates. Instead, before data is uploaded, we apply a rule-based aggregation algorithm to identify the 10 most frequently visited areas by users. These areas are represented using low-resolution area IDs, which are uploaded instead of actual latitude and longitude data, ensuring location privacy.

- **Long-term privacy protection**: These measures collectively ensure that users cannot be identified through long-term behavioral or location analysis.

All datasets are stored on secure, encrypted servers with strict access control protocols, ensuring access is granted only to authorized researchers bound by confidentiality agreements.

Furthermore, to proactively address fairness concerns, the datasets intentionally exclude any demographic or user-specific attributes, such as gender, race, or age. This design inherently reduces the risk of our models learning or perpetuating societal biases associated with these characteristics.

## REPRODUCIBILITY STATEMENT

To ensure the reproducibility of our research, we commit to making our work as transparent and accessible as possible.

[leftmargin=*]

- **Code:** The source code for our proposed model, experimental setup, and evaluation scripts will be made publicly available in a GitHub repository upon publication of this work. The repository will include detailed instructions for setting up the environment and running the experiments.
- **Implementation Details:** Key hyperparameters and architectural choices for our model are described in the main paper. A comprehensive list of all hyperparameters, along with details about the computational environment (hardware, software libraries, and versions), will be provided in the `README.md` file of our code repository.

The implementation of BUA is available online at `https://anonymous.4open.science/r/dasjijio-21B2/`

## 6 ETHICAL CONCERNS

We have taken several steps to address privacy and ethical considerations:

- **Anonymization Procedures**: Each user is assigned a random anonymous ID, which is re-generated every three months to prevent long-term tracking.
- **Location Data Handling**: We do not collect precise location coordinates. Instead, prior to data upload, we apply a rule-based aggregation algorithm that identifies the user's top 10 most frequently visited areas. These are represented using low-resolution area IDs and ultimately uploaded instead of actual latitude and longitude data, ensuring location privacy.
- **Long-Term Privacy Protection**: These measures collectively make it infeasible to identify users through behavioral or location analysis over time.

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

## A   USE OF LLMS

We used LLMs to assist in writing the paper, such as identifying typos and correcting grammatical errors, as well as polishing some paragraphs.

## B   SUMMARY OF ALL UNDERSTANDING TASKS

Summary as shown in Figure 3.

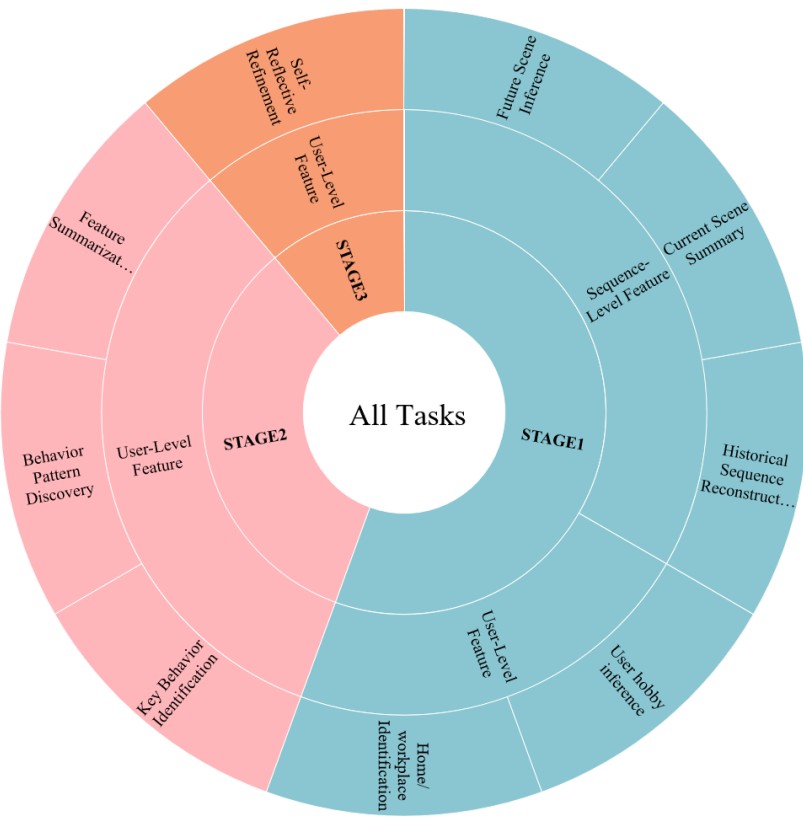

Figure 3: Summary of all understanding tasks

## C  DATASET INFORMATION

**Behavior dataset**: This large-scale dataset is derived from mobile phone usage logs. When users interact with their mobile phones, various types of logs are generated, desensitized, and reported with user consent. After desensitizing the original data, we extract 37 daily behaviors that are reliably extracted from raw logs and also cover broad life scenarios, including activities related to learning, work, entertainment, leisure, etc. The dataset spans from March 1, 2024, to April 29, 2024, and consists of over 50 million behavior events from 24,133 anonymous users. We preprocess the dataset and construct samples in the format of $(weekday, timestamp, location, behaviortype)$. Since our target is fine-tuning the LLM instead of training from scratch, we only randomly select a subset (100,000) of data for experiments.

**Tencent dataset**: The Tencent Trajectory dataset, collected from a social network, captures user mobility trajectories. Points of interest (POIs) are manually annotated with detailed intent types. The dataset includes 2,000 users and spans from October 8 to December 31, 2019, comprising 320,516 records. Each record contains an anonymized location ID, the associated intent type, and a timestamp.

The characteristics of both datasets are presented in Table 7.

Table 7: Statistics of the datasets

| Dataset | # Users | # Behav. Type | # Logs |
|---|---|---|---|
| Behavior Dataset | 24,133 | 37 | 100,000 |
| Tencent Dataset | 2,000 | 14 | 320,516 |

# D    DETAILS OF USED METRICS FOR PREDICTION

## D.1    BEHAVIOR CATEGORY

For *long-tailed learning*, following the settings of relevant work (Liu et al., 2019; Shi et al., 2024), we evaluate four accuracy metrics based on behavior occurrence frequencies: category-average accuracy across all behaviors ($Overall$), head-category accuracy for behaviors with $> 5.0\%$ frequency ($Head$), medium-category accuracy for those between 1.0% and 5.0% frequency ($Medium$), and tail-category accuracy for the remaining low-frequency behaviors ($Tail$).

## D.2    BEHAVIOR PREDICTION METRICS

The formula for $Prec_w$ :

$$Prec_w = \frac{\sum_{c \in C}(\text{TP}_c + \text{FP}_c) \cdot \text{Precision}_c}{\sum_{c \in C}(\text{TP}_c + \text{FP}_c)} \tag{6}$$

The formula for $Rec_w$ :

$$Rec_w = \frac{\sum_{c \in C}(\text{TP}_c + \text{FN}_c) \cdot \text{Recall}_c}{\sum_{c \in C}(\text{TP}_c + \text{FN}_c)} \tag{7}$$

The formula for $Overall$ :

$$Accuracy = \frac{1}{|C|} \sum_{c \in C} \frac{\text{TP}_c}{\text{TP}_c + \text{FN}_c} \tag{8}$$

The formula for $Head$ :

$$Accuary = \frac{1}{|C_h|} \sum_{c \in C_h} \frac{\text{TP}_c}{\text{TP}_c + \text{FN}_c} \tag{9}$$

The formula for $Medium$ :

$$Accuary = \frac{1}{|C_m|} \sum_{c \in C_m} \frac{\text{TP}_c}{\text{TP}_c + \text{FN}_c} \tag{10}$$

The formula for $Tail$ :

$$Accuary = \frac{1}{|C_t|} \sum_{c \in C_t} \frac{\text{TP}_c}{\text{TP}_c + \text{FN}_c} \tag{11}$$

Where $|C|$ represents the total number of classes, $|C_h|$ represents the total number of classes belonging to the head category, Where $|C_m|$ represents the total number of classes belonging to the medium category, Where $|C_h|$ represents the total number of classes belonging to the tail category. True Positives ($TP_c$) denotes the number of samples correctly classified as class $c$, False Positives ($FP_c$) represents the number of samples incorrectly classified as class $c$, and False Negatives ($FN_c$) stands for the number of samples incorrectly classified as other classes instead of class $c$. And Precision$_c$ and Recall$_c$ respectively refer to the precision and recall of class $c$.

# E    DETAILS OF USED METRICS FOR GENERATION

The formula for $BLEU$:

$$\text{BLEU} = \text{BP} \cdot \exp\left( \sum_{n=1}^{N} w_n \log p_n \right) \tag{12}$$

Where $\text{BP} = \min\left(1, e^{1-r/c}\right)$ is the brevity penalty, $p_n$ is the modified $n$-gram precision, $r$ is the reference length, and $c$ is the candidate length.

The formula for $TVD$:

$$\text{TVD}(P, Q) = \frac{1}{2} \sum_{i=1}^{k} |P(i) - Q(i)| \tag{13}$$

Where $P$ and $Q$ are probability distributions over $k$ classes, $P(i)$ denotes the predicted probability of class $i$, $Q(i)$ denotes the ground truth probability.

The formula for $JSD$:

$$\text{JSD}(P\|Q) = \sqrt{\frac{1}{2}D_{\text{KL}}(P\|M) + \frac{1}{2}D_{\text{KL}}(Q\|M)} \tag{14}$$

Where $M = \frac{1}{2}(P + Q)$ is the midpoint distribution, and $D_{\text{KL}}$ denotes the Kullback-Leibler divergence:

$$D_{\text{KL}}(P\|Q) = \sum_{i=1}^{k} P(i) \log \frac{P(i)}{Q(i)} \tag{15}$$

## F    DETAILS OF BASELINES

**SASRec (Kang & McAuley, 2018).** uses self-attention mechanisms to model user behavior sequences. It captures both short-term and long-term dependencies in sequential data, allowing it to focus on the most relevant items in the user's interaction history for recommendation.

**BehaveGPT (Gong et al., 2025)** is a transformer-based model pre-trained specifically on user behavior data. Its novel pre-training paradigm enables it to learn complex behavior patterns and support various downstream tasks, including next behavior prediction, long-term generation, and cross-domain adaptation.

**PITuning (Gong et al., 2024)** loads pre-trained Large Language Model (LLM) parameters to acquire textual knowledge and then designs an adaptive unlearning strategy to address the long-tail preference issue, achieving excellent performance in user behavior prediction.

**AlphaFuse (Hu et al., 2025)** is a simple yet effective language-guided learning strategy that addresses long-tail intent modeling by learning ID embeddings within the null space of language embeddings.

**TALLRec (Bao et al., 2023)** is one of the earlier methods to integrate Large Language Models (LLMs) with the recommendation domain. It employs a two-stage tuning process—Alpaca Tuning and Rec-Tuning—to finetune LLMs for recommendations, enabling effective and efficient adaptation of LLMs with only a small number of tuning samples.

**A-LLMRec (Kim et al., 2024)** bridges the knowledge between the language and recommendation domains by training an alignment network with a variety of tasks, targeting both warm and cold-start scenarios.

**CoLLM (Zhang et al., 2025)** captures collaboration information using external traditional models and maps it into the LLM's input embedding space as collaboration embeddings. This external integration allows effective modeling of collaboration without modifying the LLM, enabling flexible use of various collaboration modeling techniques.

**LLaRa (Liao et al., 2024)** introduces a hybrid prompting method that integrates both world knowledge and behavioral patterns into item representations. It conducts curriculum prompt tuning to achieve modality alignment.

For comparison, we also consider LLMs that are not fine-tuned on behavioral data, i.e., Deepseek-V3 (DeepSeek-AI, 2025), which is a powerful Mixture-of-Experts (MoE) language model with 671B total parameters and 37B activated per token, offering performance comparable to GPT-4 Hurst et al. (2024) at a lower cost.

## G    IMPLEMENTATION DETAILS

The hardware used in this experiment consists of 8 NVIDIA A100 40G GPUs. We selected Qwen2.5-7B (Team, 2024) as the backbone for our experiments. Our experiments utilized the AdamW optimizer with a cosine annealing learning rate schedule, setting the warm-up proportion to 0.03. The maximum learning rate for cosine annealing was set to 5e-5, while both the minimum and initial warm-up learning rates were set to 1e-6. We conducted LoRA (Hu et al., 2022) fine-tuning and parallel training acceleration. All experiments were performed with a maximum of 3 training epochs and a batch size of 96, selecting the best-performing model on the validation set for testing.

Our experiments are typically completed within 8 hours.And for the experimental results, due to limited computing resources, we fixed the random seed to 42 and only ran it once

## H    DETAILS OF PRACTICAL APPLICATIONS

### H.1    DATA GENERATION PROCESS

For the behavior dataset, we use our model and baselines to generate one day of user behavior data based on a history sequence of 100 behaviors (spanning over one day). From the generated output, we take the most recent 41 behaviors and use the first 40 to predict the final one.

### H.2    DOWNSTREAM TASK EXPERIMENTAL SETTINGS

To evaluate the utility of the generated synthetic data, we employed SASRec as the downstream behavior prediction model. To ensure a fair comparison across datasets of varying sizes (e.g., real data vs. real + synthetic data) and to address concerns regarding gradient steps, we adopted a "train to convergence" strategy. Instead of fixing the total number of gradient steps, we utilized **Early Stopping** with a patience of 5 epochs (monitoring validation loss). This approach ensures that all models, regardless of the training data volume, are trained to their maximum potential without overfitting or underfitting. The specific hyperparameters used for the downstream SASRec model are consistent with standard settings and are detailed as follows:

- **Model Architecture**:
    - Hidden Units: 50
    - Number of Blocks (Layers): 2
    - Number of Attention Heads: 1
    - Dropout Rate: 0.1
    - Max Sequence Length: 40

- **Optimization**:
    - Optimizer: Adam
    - Learning Rate: 0.001
    - Batch Size: 16
    - L2 Embedding Regularization (`l2_emb`): 0.01

- **Training Config**:
    - Maximum Epochs: 200
    - Early Stopping Patience: 5 epochs

## I    ABLATION STUDY FOR BEHAVIOR GENERATION TASK

The ablation results on the generation task are shown in the following Table 8.

Table 8: Ablation Study for Behavior Generation Task

| Method | Event | | | Timestamp | | | Location | | |
|---|---|---|---|---|---|---|---|---|---|
| | Bleu ↑ | TVD ↓ | JSD ↓ | Bleu ↑ | TVD ↓ | JSD ↓ | Bleu ↑ | TVD ↓ | JSD ↓ |
| Ours | 0.354 | 0.140 | 0.020 | 0.541 | 0.147 | 0.020 | 0.711 | 0.065 | 0.005 |
| w/o stage1 | 0.309 | 0.167 | 0.028 | 0.500 | 0.162 | 0.024 | 0.640 | 0.093 | 0.007 |
| w/o stage2 | 0.304 | 0.189 | 0.029 | 0.580 | 0.095 | 0.008 | 0.745 | 0.064 | 0.006 |
| w/o stage3 | 0.343 | 0.146 | 0.022 | 0.523 | 0.156 | 0.025 | 0.708 | 0.079 | 0.008 |

## J  ANALYSIS OF ERROR SOURCES IN BEHAVIOR PREDICTION

We additionally conducted error analysis experiments to better analyze the sources of error in the behavior prediction task. Specifically, for the three progressive subtasks in the second stage, we replaced the model-generated outputs with ground-truth values from their respective supervised training tasks. The experimental setup includes four groups, as shown below(Note that understanding task1, task2, and task3 in the table represent the User Key Behavior Identification, User Behavior Pattern, and User Feature Summarization Discovery tasks, respectively.). In the table, "pred" indicates that the corresponding feature uses the model's own generated result (which may contain errors), while "label" denotes the use of the ground-truth value from the supervised training tasks.

Table 9: Experimental Setup for Error Analysis

| ID | understanding task1 | understanding task2 | understanding task3 |
|----|---------------------|---------------------|---------------------|
| 1  | pred                | pred                | pred                |
| 2  | label               | pred                | pred                |
| 3  | label               | label               | pred                |
| 4  | label               | label               | label               |

Under these four experimental settings, we analyzed changes in the accuracy on the long-tail test set. The results are summarized in the table below. In the table, $r2w$ represents the percentage of data that changed from correct to incorrect predictions compared to the previous row's settings, while $w2r$ represents the opposite, and *Difference* indicates the net accuracy improvement (the difference between $w2r$ and $r2w$).

Table 10: Error Analysis Results on Long-Tail Test Set

| ID | $Overall$ | $r2w$ | $w2r$ | $Difference$ |
|----|-----------|-------|-------|--------------|
| 1  | 0.336     | -     | -     | -            |
| 2  | 0.417     | 6.3%  | 14.4% | 8.1%         |
| 3  | 0.432     | 2.7%  | 4.2%  | 1.5%         |
| 4  | 0.480     | 1.8%  | 6.6%  | 4.8%         |

The experimental results reveal that *User Key Behavior Identification* and *User Feature Summarization* have the greatest impact on errors. *User Key Behavior Identification* serves as the starting point for behavioral analysis in stage 2, where even small initial errors can propagate and compound across subsequent subtasks. Meanwhile, the final *User Feature Summarization* task, being directly linked to behavior prediction, significantly influences the final accuracy. The quality of the summarized features directly affects the precision of behavior predictions, hence its substantial impact.

In the paper, we primarily focused on enhancing *User Feature Summarization* through self-reflection optimization tasks. However, we acknowledge that insufficient attention was given to the *User Key Behavior Identification* task, which also has a significant impact on errors. This insight offers a valuable direction for our future work.

## K  JOINT OPTIMIZATION

**Joint Optimization** – Consider using adaptive learning rate schedules to resolve convergence mismatch between prediction and generation tasks

We implemented an **adaptive learning rate schedule** by dynamically adjusting the task loss weights based on the ratio of current loss to initial loss. This effectively assigns a higher weight to the prediction task and a lower weight to the generation task, accelerating convergence of the former while slowing down the latter. Below is a detailed description of the strategy:

### K.1  DYNAMIC TASK WEIGHTING STRATEGY

Let:

- $L_i^{(0)}$: the initial loss of task $i$

- $\hat{L}_i$: the current exponentially moving averaged (EMA) loss of task $i$

- $r_i = \frac{\hat{L}_i}{L_i^{(0)}}$: the loss ratio of task $i$

- $\bar{r} = \frac{1}{|\mathcal{V}|} \sum_{i \in \mathcal{V}} r_i$: the average loss ratio across **valid tasks**

- $s_i = \frac{\bar{r}}{r_i}$: the **relative learning speed** of task $i$ (slower tasks will have larger values)

- $\alpha$: a tunable exponent to control the sensitivity of the weighting

The normalized task weight $w_i$ is computed as:

$$w_i = \begin{cases} \frac{s_i^\alpha}{\sum_{j \in \mathcal{V}} s_j^\alpha} \cdot |\mathcal{V}|, & \text{if } L_i^{(0)} > 0 \\ 1, & \text{otherwise} \end{cases}$$

Where:

$$\mathcal{V} = \left\{ i \mid L_i^{(0)} > 0 \right\}$$

is the set of valid tasks (i.e., those with positive initial loss values).

After applying this method, the step corresponding to the lowest total loss shifted from **2600 to 3200**, with corresponding unweighted prediction and generation losses improving slightly to **0.2674** and **0.271** (from **0.2690** and **0.2726**). These data show that this method does make the convergence speed of prediction and generation tasks more matched.

The table below shows performance comparisons, where "No optimization" refers to results without multi-task optimization (as in the paper)

### K.1.1   PREDICTION TASK

Table 11: Performance of the Prediction Task Under Different Optimization Methods

| Optimization method | $Prec_w$ | $Rec_w$ | $Overall$ | $Head$ | $Medium$ | $Tail$ |
|---|---|---|---|---|---|---|
| No optimization | 0.644 | 0.642 | 0.471 | 0.538 | 0.489 | 0.446 |
| Adaptive learning rate | 0.638 | 0.648 | 0.479 | 0.528 | 0.497 | 0.452 |

### K.1.2   GENERATION TASK

Table 12: Performance of the Generation Task Under Different Optimization Methods

| Method | Event | | | Timestamp | | | Location | | |
|---|---|---|---|---|---|---|---|---|---|
| | Bleu ↑ | TVD ↓ | JSD ↓ | Bleu ↑ | TVD ↓ | JSD ↓ | Bleu ↑ | TVD ↓ | JSD ↓ |
| None | 0.354 | 0.140 | 0.020 | 0.541 | 0.147 | 0.020 | 0.711 | 0.065 | 0.005 |
| Adaptive learning rate | 0.363 | 0.141 | 0.020 | 0.553 | 0.146 | 0.019 | 0.708 | 0.079 | 0.008 |

As shown, while some metrics improved, results are not consistently better across all tasks. This suggests that **multi-task optimization requires more sophisticated strategies**, and we plan to explore further methods (e.g., separate optimizers or gradient balancing techniques) in future work.

## L   EFFICIENCY COMPARISON

**Efficiency Comparison** – Compare inference time and memory usage with baseline models

Regarding computational cost during inference:

- **Hardware:** All inference tests were conducted on NVIDIA A100 (40GB).

- **Inference Time (on 20,000 samples from the Honor dataset):**

  - *< 3 minutes:* SASRec, BehaveGPT, PITuning, AlphaFuse
  - *˜25 minutes:* TALLRec, A-LLMRec, CoLLM, LLaRa
  - *˜40 minutes:* BUA
  - *Not available:* DeepSeek (API-based)

- **Memory Usage:**

  - *Low (< 2GB):* SASRec (˜1GB), BehaveGPT, PITuning, AlphaFuse (˜2GB)
  - *High (˜30–32GB):* TALLRec, A-LLMRec, CoLLM, LLaRa, BUA

BUA's inference efficiency is comparable to other LLM-based baselines, though higher than traditional methods—reflecting a broader trend in LLM-based approaches. We anticipate continued advances in LLM optimization that will help narrow this efficiency gap in the near future.

## M    CROSS-CULTURAL CONTEXTS EVALUATION

**Cross-Cultural Contexts Evaluation** – Test the model on datasets from different domains or cultural backgrounds

To address this, we incorporated a new dataset: the **Carat Top 1000 Users App Usage Dataset**, which collects app usage and battery data from volunteers across multiple countries, including the U.S., Japan, and the U.K., and notably excludes China. This provides a complementary perspective to the Honor and Tencent datasets used in our original submission.

We compared our method (BUA) with the best-performing baselines from each category on this dataset. The results are shown in Table 13.

Table 13: Performance on Carat Top 1000 Users App Usage Dataset

| Method | $Prec_w$ | $Rec_w$ | $Overall$ | $Head$ | $Medium$ | $Tail$ |
|---|---|---|---|---|---|---|
| BehaveGPT | 0.299 | 0.318 | 0.210 | 0.303 | 0.219 | 0.052 |
| PITuning | 0.352 | 0.356 | 0.357 | 0.362 | 0.425 | 0.152 |
| CoLLM | 0.400 | 0.365 | 0.367 | 0.377 | 0.418 | 0.233 |
| Ours (BUA) | **0.447** | **0.418** | **0.400** | **0.409** | **0.451** | **0.267** |

As shown, our method continues to achieve strong performance on a dataset with a markedly different demographic and geographical distribution, further validating its generalizability.

## N    PRETRAINED BASE MODEL REPLACEMENT

**Pretrained Base Model Replacement** – Evaluate the effect of replacing the current pre-trained base model

We have replaced BehaveGPT with SASRec as the pretrained behavior sequence encoder. The performance is shown in Table 14.

Table 14: Performance Comparison of Different Pretrained Base Models

| Pretrained Model | $Prec_w$ | $Rec_w$ | $Overall$ | $Head$ | $Medium$ | $Tail$ |
|---|---|---|---|---|---|---|
| SASRec | 0.561 | 0.589 | 0.331 | 0.466 | 0.389 | 0.247 |
| BehaveGPT | 0.644 | 0.642 | 0.471 | 0.538 | 0.489 | 0.446 |

While SASRec underperforms compared to BehaveGPT, our method still achieves notable gains over SASRec alone, demonstrating its effectiveness.

## O  SELF-REFLECTION DETAILS

**Self-Reflection Details** – Provide a more detailed explanation of the self-reflection optimization method.

To clarify, the model identifies recurring shortcomings in initial profiles through prompt-guided reflection, focusing on issues such as:

1. insufficient abstract summarization,
2. inadequate detail association and reasoning,
3. poor structural clarity,
4. weak information hierarchy,
5. inaccurate temporal pattern analysis, and
6. lack of personalized expression.

The corresponding correction criteria are designed as follows:

- **For abstract summarization**: Elevate surface-level behaviors to infer deeper cognitive traits (e.g., deducing "information-driven lifestyle" from frequent news consumption).
- **For temporal analysis**: Calibrate behavior frequencies and highlight periodic patterns.
- **For structure**: Implement a three-layer hierarchy—from cognitive-level traits to habit interactions and specific behavioral anchors.
- **For personalization**: Emphasize distinctive, user-specific behavioral descriptors while avoiding vague generalities.

Importantly, this self-reflective process is not limited to output refinement. As described above, feedback from these reflections is also used to update model parameters via supervised fine-tuning, leading to further performance improvements.

## P  DATA GRANULARITY

**Data Granularity** – Clarify what is meant by "high-level daily activities" and how they are represented in the data.

To clarify the granularity of "Behavior Type ID," we define it at the level of high-level daily activities—neither raw sensor signals nor overly abstract categories. Below is a simplified example of a typical user's day to illustrate the scope:

- **Morning**: Alarm clock, check weather
- **Commute (to work)**: Subway, watch news, payment
- **Work hours**: Editing video, online meeting
- **Lunch break**: Ordering takeout, watching video
- **Commute (to home)**: Subway, watching video, payment
- **Evening**: Online shopping, gaming, watching video

Due to space limitations, this example condenses some activity details, but it reflects the typical granularity used across different scenarios.

## Q  HUMAN EVALUATION

**Human Evaluation** – Human evaluation experiments on the quality and interpretability of model-generated profile features

We have conducted a human evaluation study to more systematically assess the interpretability of the generated portrait features.

Following the methodology of prior work [1], we randomly selected 120 test samples from the Honor dataset. For each sample, portrait features were generated by three sources:

1. human annotators,

2. our proposed model (BUA),

3. the base model (Qwen2.5-7B without fine-tuning).

This resulted in a total of 360 portrait feature samples, which were evaluated by a separate group of human judges using consistent evaluation criteria.

The evaluation focused on two dimensions:

- **Rationality**: the degree to which the portrait features align with the user's historical behavior

- **Interpretability**: the extent to which the portrait features help explain the user's predicted future behavior

Scores ranged from 0 to 3, with higher scores indicating better performance. To avoid bias, the order and source of the portrait features were anonymized and randomly shuffled for each evaluation instance. (Note: If multiple sources achieve the highest score for a sample, the credit is divided equally among them.)

Table 15: Average Scores of Human Evaluation (0–3 Scale)

| Type | Rationality | Interpretability |
|---|---|---|
| Human | 2.51 | 2.55 |
| BUA | 2.46 | 2.39 |
| No Tune | 1.90 | 1.83 |

Table 16: Best Score Probability per Sample

| Category | Human | BUA | No Tune |
|---|---|---|---|
| Best in Rationality | 39.8% | 37.9% | 22.3% |
| Best in Interpretability | 45.3% | 33.3% | 21.4% |

These results indicate that while human-written features still achieve the highest overall performance, our fine-tuned model (BUA) significantly outperforms the base model in both rationality and interpretability. Moreover, BUA's performance approaches that of human-written features, demonstrating meaningful gains in interpretability.

## R  QUESTIONNAIRE SETUP DETAILS

TASK DESCRIPTION:

Your task is to evaluate the quality of user features based on two dimensions

DIMENSION 1 (RATIONALITY):

Does the user feature accurately reflect the user's historical behavior sequence?

DIMENSION 2 (INTERPRETABILITY):

Can the user feature help explain the predicted next behavior of the user? Does it provide a reasonable basis for why the predicted behavior might occur?

## Scoring Criteria for Dimension 1 (Rationality: 0–3 points)

| Score | Description |
| --- | --- |
| 0 | **No Match**: The profile feature is not reflected at all in the user's behavior sequence. |
| 1 | **Weak Match**: The profile feature is only partially reflected in the behavior sequence. |
| 2 | **Basic Match**: The feature is generally reflected in the behavior sequence but is overly broad (e.g., "user likes playing games"). |
| 3 | **Strong Match**: The feature is clearly and specifically reflected in the behavior sequence (e.g., "user likes playing games on Friday nights after watching short videos"). |

## Scoring Criteria for Dimension 2 (Interpretability: 0–3 points)

## S   Empirical Analysis of Curriculum Learning Strategy

To empirically validate the necessity and effectiveness of our proposed three-stage curriculum design, we conducted a comprehensive ablation study comparing the convergence dynamics of our approach against a standard multi-task learning baseline. Specifically, we contrasted our proposed **Staged Training** strategy, where the model is optimized sequentially through Sequence-Level Understanding (Stage 1), User-Level Feature Modeling (Stage 2), and Self-Reflective Refinement (Stage 3), against a **Joint Training** baseline. In the Joint setting, the model is trained simultaneously on all tasks across the three stages from scratch, disregarding the hierarchical dependencies inherent in behavioral understanding. This comparison aims to verify whether the structured, easy-to-hard progression provides tangible optimization benefits over simple joint optimization.

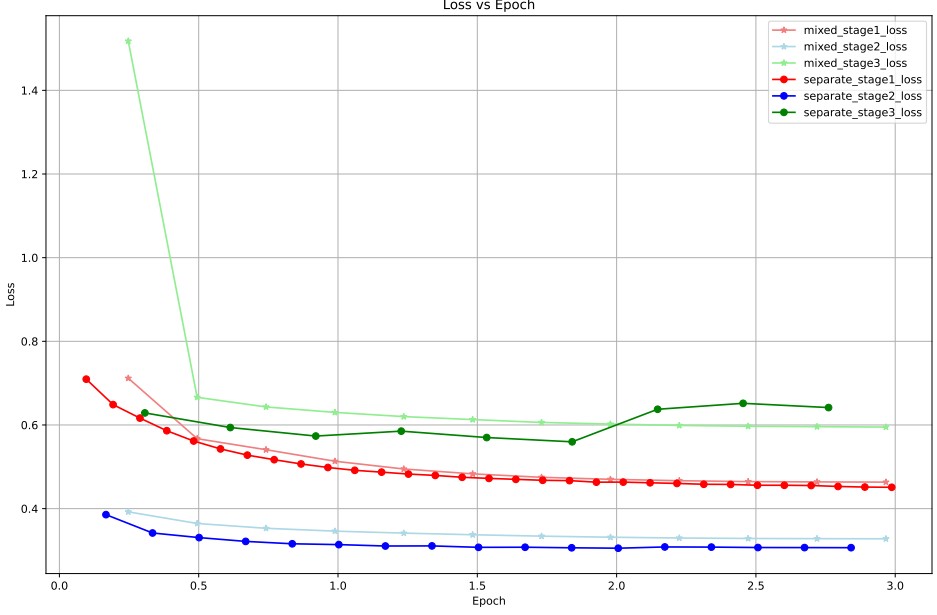

Figure 4: Validation loss comparison between our Staged Curriculum (Separate) and Joint Training (Mixed) strategies. The solid lines represent our Staged Training, while the faded lines with stars represent Joint Training.

The validation loss curves for each stage's specific tasks under both settings are presented in Figure 4. The comparative analysis reveals three critical insights regarding the training dynamics:

| Score | Description |
|-------|-------------|
| 0 | **No Match**: The profile feature is completely unrelated to the predicted user behavior. |
| 1 | **Weak Match**: The feature can be loosely connected to the predicted behavior (e.g., "user often engages in leisure activities" → predicted behavior: "playing games"). |
| 2 | **Basic Match**: The feature aligns with the predicted behavior but is too general (e.g., "user likes playing games" → predicted behavior: "playing games"). |
| 3 | **Strong Match**: The feature directly and specifically supports the predicted behavior (e.g., "user likes playing games on Friday nights after reading the news" → predicted behavior: "playing games"; it is Friday night and the user has just read the news). |

- **Comparable Performance on Baselines (Stage 1):** For the most fundamental task, Sequence-Level Understanding, the loss curves for both Separate (Red solid line) and Mixed (Red faded line) settings are relatively close. This indicates that simple semantic alignment is less sensitive to the training strategy and can be adequately learned via joint optimization.

- **Superiority in User-Level Modeling (Stage 2):** A significant divergence appears in the more difficult Stage 2 tasks. The Separate Training (Blue solid line) achieves a consistently lower minimum validation loss compared to the Mixed setting (Blue faded line). This confirms that a solid foundation in Stage 1 is essential for mastering complex user features, as the model benefits from pre-aligned semantic representations.

- **Cold-Start Challenge in Self-Reflection (Stage 3):** Notably, the Mixed_Stage3_Loss (Green faded line) starts at an extremely high value ($> 1.5$), indicating that the model struggles to perform self-reflective refinement without a pre-established user profile context. In contrast, the Staged approach (Green solid line) allows the model to tackle Stage 3 with initialized understanding, resulting in a smoother optimization landscape and lower final loss.

These empirical results strongly corroborate the theoretical foundation of our curriculum design, rooted in cognitive scaffolding and curriculum learning (Bengio et al., 2009). The convergence patterns demonstrate that while joint training is sufficient for aligning basic semantic representations, it struggles with higher-order reasoning tasks without established prerequisites. By enforcing a structured, easy-to-hard learning progression, our staged approach ensures that the model acquires a robust understanding of fundamental behavioral semantics before tackling complex user profiling and self-reflection, achieving superior performance.

