# OpenReview forum: "LLMs Reading the Rhythms of Daily Life: Aligned Understanding for Behavior Prediction and Generation"
_ICLR.cc/2026/Conference — ICLR 2026 Conference Withdrawn Submission_

### Official Review · Reviewer_uFz2 · 2025-10-26

**Soundness:** 3
**Presentation:** 3
**Contribution:** 2
**Rating:** 4
**Confidence:** 3

**Summary:**

The authors propose BUA, a training framework which bridges the modality gap between behavioral sequences and language through a three stages curriculum, sequence level understanding, user level feature modeling, and self-reflective refinement, anchored by pretrained behavior sequence embeddings. The multi-round dialogue mechanism teaches the LLM to reason through and describe behavioral patterns before predicting or generating future actions, enhancing interpretability and performance. Experiments on two real world datasets show that BUA outperforms traditional and LLM based baselines in both behavior prediction and generation, particularly for long-tail behaviors.

**Strengths:**

1. The idea of aligning behavioral embeddings with language representations via curriculum learning tackles a timely challenge in cross-modal understanding. And the structured progression from sequence understanding to reflective refinement provides a coherent training strategy and contributes to interpretability.
2. The experiments show solid gains over both traditional sequence models and LLM-based baselines, especially on long-tail behavioral patterns where previous methods struggle. So, it could be benefits to personalized assistants, recommendation systems, and behavioral analytics applications.

**Weaknesses:**

While BUA demonstrates improved performance on long-tail behaviors, the paper lacks sufficient qualitative analysis or visualization to illustrate how the model captures these rare behavioral patterns. For example, since the authors use qwen as the base model, it is possible that the observed improvements partially stem from qwen’s inherent advantages over other LLMs (e.g., DeepSeek). A comparison across different base models or an analysis isolating BUA’s contribution would help clarify this point.

**Questions:**

see weaknesses

---

> ### Author Response · Authors · 2025-11-28
> **Response  to weakness**
>
> ### **Response to Weakness: Qualitative Analysis on Long-Tail Behaviors & Base Model Comparison**
>
> We appreciate your thoughtful comment. Below, we address the concerns regarding the qualitative mechanism of long-tail modeling and the impact of the base model choice.
>
> **1. Qualitative Analysis: Capturing Semantic Prominence in Long-Tail Behaviors**
>
> We posit that our model naturally enhances long-tail behavior modeling by effectively extracting and generalizing the spatiotemporal features of users' historical patterns. Consider **subway commuting** as a typical example: although its absolute frequency is far lower than habitual behaviors like watching short videos, it signifies a major conceptual shift in user patterns, such as the transition from a "work mode" to a "home leisure mode." While this event is statistically sparse, it is **highly prominent from a behavioral semantic perspective**. This allows our "Key Behavior Identification" task in the understanding phase to explicitly detect and highlight it. Subsequently, the model further analyzes and summarizes the specific timing and characteristics of this commuting habit based on the settings of the understanding task. Furthermore, leveraging our multi-turn dialogue framework, the model can recall these pre-generalized features during the prediction phase. When the prediction context matches this commuting pattern, the LLM prioritizes the behavior based on the user’s logical daily rhythm rather than raw historical frequency, thereby significantly improving prediction accuracy for such rare but semantically crucial events.
>
>
>
> **2. Base Model Comparison: Improvements Stem from BUA, Not Qwen**
>
> Regarding the concern about the base model, we can confirm that the observed improvements stem from our BUA framework rather than the inherent advantages of Qwen. To ensure a strictly fair comparison, we implemented all LLM-based fine-tuning baselines (including TALLRec, A-LLMRec, CoLLM, and LLaRA) using the exact same backbone model Qwen2.5-7B in our paper; therefore, the significant performance gap is entirely attributable to our alignment strategy. Moreover, we compared our method against DeepSeek-V3, a massive Mixture-of-Experts model with 671B parameters, and found that our fine-tuned 7B model significantly outperforms this state-of-the-art general LLM on domain-specific tasks. Finally, as shown in the table below, the raw Qwen2.5-NT (No Tune) model performs poorly without our method, further isolating the substantial contribution of our proposed framework.
>
> | method  | $Rec_w$ | $Prec_w$ |$Overall$ | $Head$|$Medium$ |$Tail$ |
> |  -------- | -------- | -------- | -------- | -------- |-------- |-------- |
> | qwen2.5-NT |0.302  |  0.425  | 0.168   | 0.228   |0.214 |0.149 |
> | Deepseek-V3 | 0.492 | 0.495 | 0.237 | 0.330 | 0.265 | 0.191 |
> | TALLRec | 0.617 | 0.607 | 0.398 | 0.452 | 0.434 | 0.355 |
> | A-LLMRec | 0.584 | 0.557 | 0.348 | 0.422 | 0.394 | 0.299 |
> | CoLLM | 0.618 | 0.596 | 0.408 | 0.453 | 0.448 | 0.363 |
> | LLaRA | 0.615 | 0.608 | 0.404 | 0.462 | 0.439 | 0.361 |
> | **BUA** | **0.644** | **0.642** | **0.471** | **0.538** | **0.489** | **0.446** |
>
>
> Finally, thank you again for your valuable suggestions. We hope our response has addressed your concerns. If you have any further questions, we look forward to further communication.

---

### Official Review · Reviewer_g36F · 2025-10-29

**Soundness:** 2
**Presentation:** 3
**Contribution:** 2
**Rating:** 2
**Confidence:** 3

**Summary:**

In this work, the Behavioral Understanding Alignment (BUA) framework is introduced in an attempt to understand human daily behavioral sequences. It combines a three-stage curriculum learning pipeline with dialogue mechanisms. Experiments show that BUA performs well in prediction and generation tasks.

**Strengths:**

- Explanations are well-organized and straightforward.
- I like the different types of experiments the authors tried out. Especially, the cross-model enhancement experiment is informative.

**Weaknesses:**

### 1. Task Clarification
I am not fully convinced about the difference between the two tasks: (1) behavior prediction and (2) behavior generation. To me, generation seems to be a sequential prediction problem that inherently depends on the prediction task. Could the authors clarify what conceptual or methodological differences I might be missing here?
### 2. Fairness in Comparison
The comparison does not appear fully fair. Most baselines are either pretrained for general purposes or trained on unrelated tasks, while BUA is explicitly trained on the downstream task. Wouldn’t it be more appropriate to compare BUA’s results with fine-tuned versions of the baselines under equivalent conditions?
### 3. Curriculum Design and Theoretical Grounding
The three-stage curriculum (sequence-level, user-level, self-reflective) lacks theoretical grounding and is mostly procedural. Please provide a convergence curve proving that the staged training outperforms simple joint training beyond anecdotal evidence.
### 4. Evaluation Metrics and Interpretability Claims
Metrics such as weighted precision/recall, BLEU, TVD, and JSD are low-level and insufficient for the paper’s conceptual claims about understanding or interpretability. If this is a standard metric for the task, I would suggest toning down the conceptual claims. Similarly, the authors claim that interpretability is a feature of BUA, but only a few qualitative examples are provided. How interpretability or explainability is enhanced, compared to previous works, is a bit vague to me.
### 5. Syntehtic Data Experimental Details
In the synthetic data experiment, I think more experimental details are needed. Are "Real + Ours" trained with a commensurable number of gradient steps compared to "Real"? If not, please provide results where you control the number of epochs to make total number of gradient steps similar between the two setups. Also, please provide other experimental details somewhere in the appendix as well.
### 6. Minor Points
- BehaviorGPT --> BehaveGPT
- I suggest breaking Section 4 down into two sections (e.g. Experiments / Analysis).

**Questions:**

See Weaknesses

---

> ### Author Response · Authors · 2025-11-28
> **Response-Part 1**
>
> ### **Response to Weakness-1: Task Clarification (Prediction vs. Generation)**
>
> We appreciate your comment. While behavior generation can be viewed as a multi-step sequence prediction from a very broad technical perspective, we emphasize that they differ fundamentally in application scenarios, optimization goals, and evaluation metrics. We clarify these distinctions below:
>
> **1. Divergence in Scenarios and Objectives:**
> **Behavior Prediction** focuses on short-term capability, primarily serving intelligent user recommendation systems. The core requirement is to accurately predict the immediate next behavior based on historical context to maximize service relevance. In contrast, **Behavior Generation** targets long-term modeling capabilities. Its primary application scenario is privacy-preserving data synthesis—generating large volumes of synthetic behavior sequences for downstream tasks (e.g., training prediction models). Unlike prediction, generation does not require the output sequence to be exactly identical to the ground truth; instead, it demands that the *distribution* of the generated sequences statistically resembles the real data distribution.
>
> **2. Empirical Evidence of Task Distinction:**
> These two tasks are not simply equivalent, and success in one does not guarantee success in the other. An intuitive proof of this distinction is that a high-performing generation model does not necessarily function as a superior prediction model, and vice versa. This separation is further evidenced by the disparity in the research landscape observed in our baseline selection: there is an extensive body of related work for the prediction task (e.g., SASRec, AlphaFuse, TALLRec,A-LLMRec,LLara), whereas related work for long-term behavior generation is relatively scarce (e.g., SAND, D2A). This indicates that the research community treats them as distinct problems with unique challenges, rather than interchangeable tasks.
>
> ### **Response to Weakness-2: Fairness in Comparison**
>
> We respectfully clarify that there may be a misunderstanding regarding our experimental setup. We assure that **all experiments were conducted under strictly fair and equivalent conditions**. We address this in two parts:
>
> **1. Equivalent Training Data and Objectives for Baselines:**
> Contrary to the concern that baselines were merely pre-trained or trained on unrelated tasks, **all baseline models** (including traditional methods like SASRec and LLM-based methods like TALLRec, LLaRa, and CoLLM, with the sole exception of DeepSeek) were **explicitly trained or fine-tuned on the exact same downstream training dataset** as BUA.
> *   **Data Parity:** Although input formats differ (e.g., numerical sequences for SASRec vs. converted natural language text for TALLRec), the underlying data content and sample size used for training were identical across all models.
> *   **Task Parity:** All models were optimized for the specific downstream tasks (Behavior Prediction or Generation) reported in the paper. We did not evaluate them on "unrelated tasks," nor did we rely solely on their pre-trained weights without task-specific adaptation. Therefore, BUA is indeed being compared against the **fine-tuned versions** of these baselines.
>
> **2. Standard Practice for Large-Scale Models (DeepSeek):**
> The only exception is **DeepSeek**. Due to its massive parameter size (671B), fine-tuning it is computationally infeasible for typical academic research environments. Evaluating such ultra-large-scale LLMs (like GPT or DeepSeek) in a zero-shot or few-shot setting via API is a **standard practice** in the recommender system and user modeling community, as seen in related works [1, 2]. This serves as a reference point for the capabilities of general-purpose SOTA models compared to our specialized, fine-tuned framework.
>
> **References:**
> [1] Bao, Keqin, et al. "Tallrec: An effective and efficient tuning framework to align large language model with recommendation." Proceedings of the 17th ACM conference on recommender systems. 2023.
> [2] Liu, J., et al. "Is chatgpt a good recommender? A preliminary study. arxiv 2023." arxiv preprint arxiv:2304.10149.

---

> > ### Author Response · Authors · 2025-11-28
> > **Response-Part 2**
> >
> > ### **Response to Weakness-3: Curriculum Design and Theoretical Grounding**
> >
> > We appreciate your request for deeper theoretical and empirical justification. We have addressed this by adding a Convergence Analysis section in Appendix S.
> >
> > 1. Theoretical Grounding: Cognitive Scaffolding & Hierarchical Dependency
> >
> > Our three-stage curriculum is grounded in the Scaffolding Theory from cognitive science and Curriculum Learning in machine learning[1].
> > *   Hierarchical Dependency: Behavioral understanding possesses an inherent hierarchy: the model must first understand *what* happened (Stage 1: Sequence Semantics) before it can deduce *who* did it (Stage 2: User Patterns), and finally *refine* that understanding (Stage 3). Our staged approach ensures that the model has the necessary semantic "preconditions" before processing higher-order reasoning, ensuring that the model can handle higher-order reasoning tasks more effectively, rather than trying to learn everything simultaneously from scratch.
> >
> >
> > 2. Empirical Proof (Convergence Analysis):
> >
> > As shown in the newly added **Figure 4 in Appendix S** (please download and view our updated paper as images cannot be pasted in replies), we can observe the following:
> > *   Superiority on Complex Tasks: While performance on the simpler Stage 1 is comparable, Staged Training (Solid lines) consistently achieves lower validation loss on the more complex Stage 2 and Stage 3 tasks compared to Joint Training (Faded lines).
> > *   Stage 3 Cold-Start: Notably, in Joint Training, the Stage 3 loss (Faded Green) starts at an extremely high value (>1.5), indicating the model struggles to perform self-reflection without prior context. In contrast, our staged approach enables a smoother transition and better convergence for this advanced task.
> >
> > **References:**
> > [1] Bengio, Yoshua, et al. "Curriculum learning." Proceedings of the 26th annual international conference on machine learning. 2009.
> >
> >
> > ### **Response to Weakness-4: Evaluation Metrics and Interpretability Claims**
> >
> > We appreciate your feedback and wish to clarify the specific scope and purpose of the different metrics used in this paper to dispel any misunderstandings.
> >
> > In our experimental setup, metrics such as weighted precision and recall were used to evaluate the accuracy of the behavior prediction task, while BLEU, TVD, and JSD were specifically used to measure the distributional similarity between generated sequences and real data in the behavior generation task, and did not directly measure the interpretability of the model.
> >
> > To more accurately evaluate the interpretability of the model, we followed the work on behavior interpretability [1] and conducted a dedicated human evaluation, the details of which are provided in Appendix Q. This experiment quantitatively evaluated the generated user features based on “reasonableness” and “interpretability,” showing that BUA significantly improved interpretability compared to the base model. In addition, we also provide qualitative examples in the paper to support this assessment (Section 4.7).
> >
> > We recognize that placing this key evaluation in the appendix may be misleading, giving the impression that our conclusions rely solely on qualitative cases. Therefore, in the final version, we will move the quantitative human evaluation results from Appendix Q to the main experimental section to more clearly demonstrate our conclusions regarding interpretability.
> >
> >
> > [1] Lei, Yuxuan, et al. "Recexplainer: Aligning large language models for explaining recommendation models." Proceedings of the 30th ACM SIGKDD Conference on Knowledge Discovery and Data Mining. 2024.

---

> ### Author Response · Authors · 2025-11-28
> **Response-Part 3**
>
> ### **Response to Weakness-5: Synthetic Data Experimental Details**
>
> Thank your for the careful examination of our experimental setup. We acknowledge the concern regarding the number of gradient steps. Below, we clarify our training strategy, introduce a new control experiment to validate our claims, and provide the detailed hyperparameters used.
>
> **1. Clarification on Training Strategy (Convergence vs. Fixed Steps)**
>
> The primary goal of this experiment is to verify the effectiveness of the generated data when used for augmentation—specifically, to verify whether the data generated by BUA can enhance the performance of downstream models. Therefore, it is more reasonable not to fix the number of gradient steps (which may lead to underfitting of models trained on large datasets), but to adopt a "train to convergence" strategy, which is also the standard practice in data augmentation and synthetic data research[1]. Therefore, for all settings (Real, Real + BUA, etc.) in our paper, we trained the downstream model until the validation loss stopped improving (using Early Stopping with a patience of 5 epochs). This ensures that each model reaches its optimal performance capability, providing a fair comparison of the *data quality's impact* rather than the training duration.
>
> **2. New Control Experiment: Real + Sample**
>
> To further address the concern that performance gains might stem simply from "seeing more samples" or "more gradient updates," we conducted an additional control experiment: **"Real + Sample"**.
> *   **Setup:** We constructed a baseline where we augmented the real data with sequences randomly sampled from the real data distribution (matching the statistical distribution of the original dataset but lacking the learned sequential logic of BUA).
>
> As shown in the table below, simply adding this sampled data significantly degrades performance compared to the "Real Only" baseline. Furthermore, while other generation methods like SAND ("Real Data + SAND") provide some improvement, the magnitude is substantially lower than that of our method. Ultimately, the fact that "Real + BUA" significantly outperforms both the "Real + Sample" control and the original baseline confirms that the performance boost is derived from the high-quality, semantic-rich patterns generated by BUA, rather than merely from increased gradient updates or data volume.
>
> **Table1: Comparison of Augmentation Strategies across Different Real Data Sizes**
>
> | Real Data Size | Method | Gen. Data Size | **Accuracy** |
> | :--- | :--- | :--- | :--- |
> | 500 | Real Only | 0| 0.1551 |
> | 500 | SAND | 2000 | 0.1823 |
> | 500 | BUA (Ours)| 2000 | **0.1989** |
> | 500 | Sample | 2000 | 0.1349 |
> | | | | |
> | 1000 | Real Only | 0 | 0.1969 |
> | 1000 | SAND | 2000 | 0.2105 |
> | 1000 | BUA (Ours) | 2000 | **0.2216** |
> | 1000 | Sample | 2000 | 0.1488 |
> | | | | |
> | 2000 | Real Only | 0 | 0.2359 |
> | 2000 | SAND | 2000 | 0.2561 |
> | 2000 | BUA (Ours)| 2000 | **0.2655** |
> | 2000 | Sample | 2000 | 0.1587 |
>
> **3. Detailed Experimental Hyperparameters**
>
> As requested, we have included the detailed hyperparameters for the downstream model (SASRec) used in this experiment. These details have also been added to **Appendix H**.
> *   **Optimizer:** Adam (`lr=0.001`, `beta1=0.9`, `beta2=0.98`)
> *   **Batch Size:** 16
> *   **Hidden Units:** 50
> *   **Number of Blocks (Layers):** 2
> *   **Number of Attention Heads:** 1
> *   **Dropout Rate:** 0.1
> *   **Max Sequence Length:** 40
> *   **Training Epochs:** Max 200 (with Early Stopping-5 epoch)
>
> [1] Yuan, Yuan, et al. "Learning to simulate daily activities via modeling dynamic human needs." Proceedings of the ACM Web Conference 2023. 2023.
> ### **Response to Weakness-6: Minor Points**
> Thank you for the attention to detail and the constructive suggestions regarding the manuscript's presentation.
> 1. Correction of Model Name
> We apologize for the error. We have corrected all instances of "BehaviorGPT" to "BehaveGPT" throughout the revised manuscript.
> 2. Structure of Section 4
> We appreciate the suggestion to split Section 4 into separate "Experiments" and "Analysis" sections. However, after careful consideration, we have decided to maintain the current structure where the analysis immediately follows the presentation of each sub-experiment's results. We believe this format follows common conventions in this field and allows readers to directly connect the quantitative evidence with our interpretation without needing to jump between sections.
> That said, we find your suggestion very constructive in highlighting the need for clarity. We have carefully revised the writing and transitions within Section 4 to ensure the narrative flows more naturally.
>
> Finally, thank you again for your valuable suggestions. We hope our response has addressed your concerns. If you have any further questions, we look forward to further communication.

---

### Official Review · Reviewer_D24p · 2025-10-30

**Soundness:** 3
**Presentation:** 2
**Contribution:** 2
**Rating:** 4
**Confidence:** 3

**Summary:**

This paper proposes Behavior Understanding Alignment (BUA) that integrates LLMs into human behavior modeling. BUA employs sequence embeddings from pretrained behavior models as alignment anchors and guides the LLM through a three-stage curriculum. Experiments on two real-world datasets demonstrate the effectiveness of BUA.

**Strengths:**

1, This paper enhances the LLM to understand human daily behavior sequences, which is helpful in many downstream tasks.
2. The three-stage curriculum learning pipeline works in aligning two models
3. Experimental results on two real-world datasets demonstrate the effectiveness of BUA method.

**Weaknesses:**

1, It should discuss more design choices to handle the behavior data.
2. The model design and similar training strategy can be found in different works
3. The presentation can be improved.

**Questions:**

1.	The human behavior data have the large chances to be in the training corpus. Thus, the LLM can understanding the meaning. Even if the data is collected in a structure form, the data can be translated into the natural language with the explanation of each symbols. Such a method can be more flexible in handling the dynamic data collected. I guess that the LLM method in Table accept the data in the raw form.

2.	The challenges faced by this paper, including long-tail behaviors and human-interpretable reasoning, also exist in other application domains, like adopting LLM into the medical or finance domain. It is better to include the methods in other domain and discuss the differences among them.

3.	In other part of the paper, the major challenges become the ID and embedding, e.g., the statement “a critical modality gap exists: human behavioral data, typically represented as sequences of IDs or embeddings”. Then, there is some kinds of confusion, like the relationship to the long tail mentioned early?

4.	You introduce another model when the user behavior data is too long to be put into the context. In such a context, one choice is to use LLM to summarize the data in the text form, without needing the alignment between two models. Such a method can be found in the agent memory, which also focuses on the user modeling. It is better to include them in the paper.

5.	The model framework is commonly used to handle multiple-modality problem, and the strategy to train/frozen different parts in a large model is also used in other work

6.	Figure 1 is not easy to understand. And I cannot see the challenges mentioned in the paper in this example. The features in the Figure 1(b) had better be explained more.


7.	When we fine-tune the model to understand the embedding from another sequence model, the order and kinds of tokens in the sequence model are important. How about the data format changes in the sequence model? In such a case, the embedding outputted from the sequence model may be changed, and the alignment may be invalid.

---

> ### Author Response · Authors · 2025-11-28
> **Response-Part 1**
>
> Since your **"Question"** already includes the issue described in **"Weakness"**, we have not provided a separate response specifically addressing "weakness" to avoid duplicate replies. We appreciate your understanding.
>
> ### **Response to Question-1: Input Formats and Text vs. Embedding Alignment**
>
> We appreciate your comment and would like to clarify our input settings and the rationale behind our hybrid modeling approach:
>
> 1.  **Clarification on Input Formats:**
>     We clarify that our BUA framework employs a hybrid input format, accepting both text data converted to natural language and raw sequence data encoded by a pre-trained encoder. This distinguishes our approach from the LLM-based baseline methods in Table 1 (e.g., TALLRec[1], A-LLMRec[2], LLaRa[3], and CoLLM[4]), which primarily rely on converted natural language text as input. Notably, while methods like A-LLMRec and CoLLM support *item-level* ID embeddings from pre-trained encoders, our model uniquely integrates *sequence-level* encoded vectors to more comprehensively represent the user's behavioral history.
>
> 2.  **Necessity of Hybrid Modeling:**
>     As discussed in our related work, while transformed natural language text enables LLM models to leverage their semantic understanding capabilities, traditional models possess unique advantages in capturing the inherent collaborative knowledge and complex sequence patterns within the raw data. By combining natural language input with behavioral embeddings from traditional sequence models, our approach coordinates the semantic reasoning of the LLM model with the collaborative signals from the encoder, thereby comprehensively enhancing modeling capabilities. The necessity of combining textual semantics with collaborative representations from traditional models has also been discussed in recent literature such as A-LLMRec[2] and CoLLM[4].
>
> [1] Bao, Keqin, et al. "Tallrec: An effective and efficient tuning framework to align large language model with recommendation." Proceedings of the 17th ACM conference on recommender systems. 2023.
>
> [2] Kim, Sein, et al. "Large language models meet collaborative filtering: An efficient all-round llm-based recommender system." Proceedings of the 30th ACM SIGKDD Conference on Knowledge Discovery and Data Mining. 2024.
>
> [3] Liao, Jiayi, et al. "Llara: Large language-recommendation assistant." Proceedings of the 47th International ACM SIGIR Conference on Research and Development in Information Retrieval. 2024.
>
> [4] Zhang, Yang, et al. "Collm: Integrating collaborative embeddings into large language models for recommendation." IEEE Transactions on Knowledge and Data Engineering (2025).

---

> > ### Author Response · Authors · 2025-11-28
> > **Response-Part 2**
> >
> > ### **Response to Question-2: Comparison with Other Domains (Medical/Finance)**
> >
> > We appreciate your suggestion to explore our work within a broader domain context. We address the connections and distinctions as follows:
> >
> > **1. Domain Differences and Adaptation Logic:**
> > We acknowledge that handling long-tail distributions and requiring interpretable reasoning are shared challenges across high-stakes domains like healthcare and finance. However, the underlying **data characteristics** and **reasoning logics** diverge significantly, preventing the direct transfer of methods. Medical and financial applications primarily focus on aligning LLMs with **objective expert knowledge** and rigid rules, such as diagnosing based on pathology or forecasting based on market laws. In contrast, human behavior modeling necessitates understanding **subjective user intentions** that are inextricably linked with dynamic **spatiotemporal contexts**. This distinction dictates the solution for long-tail issues: while medicine often relies on **Retrieval-Augmented Generation (RAG)**[1] to access external expert knowledge bases for rare diseases, human behavior modeling lacks a universal "encyclopedia" for private individuals.
> >
> > **2. Reflection of Cross-Domain Paradigms in Baselines:**
> > Despite these domain differences, we ensure that the core algorithmic paradigms successfully applied in these domains are effectively reflected in our chosen baseline models. Broadly speaking, the pure instruction tuning approach used in models like **Med-PaLM**[2] to infuse domain knowledge is embodied in our baseline model TALLRec, which leverages cues to align a Large Language Model with behavioral data. Similarly, the heterogeneous data source integration concept employed in financial models like **FinGPT**[3] —combining numerical signals (e.g., stock prices) with textual data (e.g., news)—is also reflected in baseline models such as CoLLM, which aim to address the same fundamental challenge of fusing heterogeneous signals (collaborative embeddings and text) to enhance the modeling capabilities of Large Language Models. By benchmarking against these representative methods, our experiments implicitly evaluate the effectiveness of these cross-domain approaches in the specific context of behavioral modeling, further validating the superiority of our proposed framework.
> >
> > [1] Zakka, Cyril, et al. "Almanac—retrieval-augmented language models for clinical medicine." Nejm ai 1.2 (2024): AIoa2300068.
> >
> > [2] Singhal, Karan, et al. "Large language models encode clinical knowledge." Nature 620.7972 (2023): 172-180.
> >
> > [3] Liu, Xiao-Yang, et al. "Fingpt: Democratizing internet-scale data for financial large language models." arxiv preprint arxiv:2307.10485 (2023).
> >
> > ### **Response to Question-3: Relationship between Long-Tail and Modality Gap**
> >
> > Thank you for highlighting this point, and we would like to clarify the logical hierarchy between these two challenges. There is no confusion or contradiction; rather, they form a **progressive cause-and-effect relationship** that drives our research motivation.
> >
> > **The logic unfolds as follows:**
> > The **long-tail distribution** represents the fundamental **application-level challenge** in behavior modeling, where traditional models fail due to data sparsity. To address this, we aim to leverage the semantic generalization and reasoning capabilities of LLMs, which are ideal for inferring low-frequency behaviors. However, this solution strategy encounters a specific **technical implementation challenge**: a critical **modality gap** exists because LLMs are trained on natural language, whereas human behavioral data is represented as **Numeric IDs**. Therefore, bridging this modality gap is the prerequisite technical step required to unlock the LLM's potential for solving the long-tail problem. In short, the "modality gap" is the specific technical barrier we must overcome to achieve the broader goal of "long-tail modeling."

---

> > > ### Author Response · Authors · 2025-11-28
> > > **Response-Part 3**
> > >
> > > ### **Response to Question-4: Model Architecture and Agent Memory**
> > >
> > > We appreciate your comment and would like to clarify our architectural design while discussing the potential integration of Agent Memory:
> > >
> > > **1. Clarification on Architecture and Input Consistency:**
> > > First, we respectfully clarify that our framework **does not** introduce an additional model conditionally based on data length. The sequence encoder (e.g., BehaviorGPT) mentioned in the paper is an **integral, permanent component** of the BUA framework. For **all** input data, regardless of sequence length, our model consistently employs a **hybrid input strategy**: it simultaneously ingests the sequence encoding vector (extracted by the pre-trained encoder to capture latent collaborative patterns) and the converted natural language text. The sequence encoder serves as the foundational anchor for modality alignment, not as a fallback module.
> > >
> > > **2. Discussion and Adoption of Agent Memory:**
> > > We agree that the "agent memory" mechanism—utilizing LLM to summarize and store historical information—is a valuable direction for user modeling, and we commit to discussing this paradigm specifically in the revised draft. In the future, we may consider integrating this concept into our framework. We will implement a **hierarchical semantic memory** module that will leverage the user feature understanding capabilities developed in the second phase. Specifically, the model can be designed to periodically generate text summaries of past behavioral fragments, thus forming a long-term memory. When reasoning about long sequences, the system will retrieve relevant high-level summaries based on the current context, rather than processing the entire original historical record, thereby effectively extending the model's sequence modeling capabilities.
> > >
> > > ### **Response to Question-5: Novelty of Framework and Training Strategy**
> > >
> > > We appreciate your comments and frankly acknowledge that our model architecture (utilizing projection layers to map the encoder's sequence encoding vectors to the LLM's feature vector space) draws inspiration from some successful paradigms in the field of visual multimodal learning, such as LLaVA[1]. And the strategy of freezing specific model components is indeed a widely used technique in the industry to accelerate training, aiming to balance training efficiency and performance, but this is not our claimed core contribution. **Our contribution lies in proposing a Behavior Understanding Alignment (BUA) framework specifically tailored to human behavior**.
> > >
> > > Unlike common visually semantically explicit image-text alignment, human behavior involves implicit intentions and abstract patterns hidden in ID sequences. To bridge this unique modal difference, our core contributions are reflected in the following two aspects:
> > >
> > > 1. We are the first to propose training a Language Learning Model (LLM) to explicitly understand sequences of everyday human behavior—through the alignment of behavioral and linguistic modalities—as a foundational step to improve downstream prediction and generation tasks.
> > >
> > > 2. We designed a domain-specific three-stage curriculum (sequence-level understanding → user-level feature modeling → self-reflection and improvement), which gradually guides LLM from capturing surface-level transformations to inferring deeper user intentions. Additionally, we introduced a multi-turn dialogue mechanism combined with a weighted loss strategy, enabling the model to perform tasks such as interpretable reasoning, behavior prediction, and future sequence generation within a single, unified framework.
> > >
> > > [1] Liu, Haotian, et al. "Visual instruction tuning." Advances in neural information processing systems 36 (2023): 34892-34916.

---

> > > > ### Author Response · Authors · 2025-11-28
> > > > **Response-Part 4**
> > > >
> > > > ### **Response to Question-6: Clarity and Implications of Figure 1**
> > > >
> > > > We appreciate your feedback. Figure 1 is designed to provide a comprehensive roadmap of how our framework systematically addresses the core challenges (Modality Gap, Long-Tail, and Interpretability). We offer a detailed breakdown of the three panels below to clarify the logical flow and feature definitions:
> > > >
> > > > **1. Figure 1(a): Structural Modality Alignment**
> > > > This panel illustrates the **foundational architecture** of BUA. By employing a hybrid input strategy, we feed the sequence encoding vector (from the frozen encoder) into the **Projection Layer**.
> > > > *   **Addressing the Challenge:** This explicitly solves the **Modality Gap at the dimensional level**. The projection layer maps the dense, numerical vector space of the behavior encoder into the semantic feature space of the LLM, establishing the structural basis for alignment.
> > > >
> > > > **2. Figure 1(b): Semantic Modality Alignment & Long-Tail Modeling**
> > > > This panel visualizes our **Three-Stage Curriculum Learning** strategy (sequence-level understanding → user-level feature modeling → self-reflection and improvement).
> > > > *   **`seq-fea` (Sequence-Level Features):** Represents **Stage 1**, where the model learns to reconstruct basic sequence information (e.g., location, time) from the embeddings.
> > > > *   **`user-fea` (User-Level Features):** Represents **Stage 2**, where the model extracts high-level user traits and patterns (e.g., "commuter habits") from the sequence.
> > > > *   **`refined-fea` (Refined Features):** Represents **Stage 3**, where the model performs self-reflection to correct and optimize the user profile.
> > > > *   **The Check/Cross Marks ($\checkmark/\times$):** These indicate whether the features learned by the model are correct.
> > > > *   **Addressing the Challenges:** By progressively learning these correct user features, the model is able to capture users' latent intentions. It addresses the modality gap problem at a **deep semantic level**, and this semantic feature understanding is key to solving the **long-tail behavior** problem because it enables the model to infer rare behaviors based on robust user profiles (not just ID co-occurrence). Additionally, the generated user features also **enhance the model's interpretability**.
> > > >
> > > > **3. Figure 1(c): Multi-Tasking & Chain-of-Thought**
> > > > This panel depicts the **Multi-Round Dialogue** setting equipped with a **Loss Balancing Strategy**.
> > > > *   **Addressing the Challenge:** This setup enables the model to simultaneously handle **Behavior Understanding, Prediction, and Generation**, making it multifunctional.
> > > >
> > > > ### **Response to Question-7: Impact of Data Format Changes on Alignment**
> > > >
> > > > We appreciate your inquiry regarding the robustness of alignment under data changes. We address this by discussing two distinct scenarios based on the degree of change:
> > > >
> > > > **1. Stability of High-Level Behaviors in Real-World Scenarios:**
> > > > In our research, the "data format" of behavior types is relatively stable. Unlike e-commerce products that are updated hourly, the high-level daily activities we model (e.g., *eating, commuting, sleeping*) represent basic human daily activities and do not undergo drastic short-term changes. Therefore, the behavior label set remains consistent over time. As for changes in the "label order" (in fact, the sequence after the change in order can be viewed as a behavior sequence of another user), as long as the behavior types remain within the original distribution range, the pre-trained sequence encoder can generate effective embedding vectors, and the alignment projection layer can correctly map them to the LLM space, thus ensuring the robustness of the framework to changes in daily behaviors.
> > > >
> > > > **2. Handling Dramatic Pattern Changes:**
> > > > In extreme cases, if the "label type" changes significantly (e.g., introducing a large number of entirely new behavior categories), this will fundamentally change the feature space of the encoder, effectively constituting a new dataset. In this scenario, it is indeed necessary to retrain the projection layer to realign the new embedding space with the LLM. We note here that this is an **inherent characteristic** of all projection-based modal alignment frameworks (such as LLaVA or CoLLM) – the alignment adapter must be updated if the feature space of the upstream encoder changes significantly. This paper focuses on constructing the alignment framework itself and therefore does not discuss challenges such as "open set recognition" in sequence encoders.
> > > >
> > > >
> > > > Finally, thank you again for your valuable suggestions. We hope our response has addressed your concerns. If you have any further questions, we look forward to further communication.

---

### Official Review · Reviewer_9zNM · 2025-11-01

**Soundness:** 3
**Presentation:** 3
**Contribution:** 3
**Rating:** 6
**Confidence:** 4

**Summary:**

In this paper, the authors propose to integrate large language models (LLMs) into human daily behavior modeling and represent human behaviors as event sequence.

The proposed framework BUA leverages sequence embeddings from BehaviorGPT as anchors to bridge the modality gap between behavioral data and natural language through a three-stage curriculum.

The experiments are conducted on two real-world user behavior datasets. The experimental results demonstrate improvements in both weighted precision and recall in prediction when it also outperforms certain baselines like SAND and D2A in generation tasks. The authors also conduct ablation studies to confirm that curriculum and multi-round dialogue settings are crucial.

**Strengths:**

* BUA effectively bridges the modality gap between behavioral sequences and LLMs through curriculum learning and sequence embeddings, enabling unified handling of prediction and generation while improving long-tail performance.
* This work demonstrates significant improvements over diverse baselines on real-world datasets, with detailed ablations, error analyses, and cross-model enhancements highlighting robustness and practical utility.
* The multi-round dialogue and self-reflective refinement produce human-readable textual summaries with better interpretability.

**Weaknesses:**

* Performance relies heavily on strong pre-trained behavior models like BehaviorGPT. The ablation study also shows drops when using weaker alternatives.
* The granularity and scope of activity modeling are limited. The study focuses on high-level daily activities and potentially overlooks nuanced behaviors.

**Questions:**

* How does the BUA framework handle missing or incomplete behavioral data in real-world scenarios? I wonder how the framework would deal the gaps or inconsistencies in behavioral sequences, which are much common in real-world logs and trajectory datasets.
* How does the BUA framework handle multi-user or group behavior modeling? The paper focuses on individual user behavior sequences but does not discuss extending the model to capture interactions or collective behaviors among multiple users.
* What are the limitations of freezing specific model components during different training stages? The paper mentions freezing certain parameters for efficiency, but it does not discuss potential trade-offs or constraints this imposes on model adaptability.

**Details Of Ethics Concerns:**

N/A.

---

> ### Author Response · Authors · 2025-11-28
> **Response-Part 1**
>
> ### **Response to Weakness-1: Reliance on Strong Pre-trained Models**
>
> We appreciate your comment. We believe this relevance reflects the **effectiveness and scalability** of our alignment framework, rather than its limitations. We explain this from three aspects:
>
> 1. **Improved Generality**: BUA acts as a general performance amplifier. Although its absolute performance is lower than BehaviorGPT when using **SASRec**, BUA still significantly improves the performance of the SASRec backbone model (raising $Rec_w$ from 0.546 to **0.589**, a **7.9% improvement**). This demonstrates the generality of our approach, unlocking the potential of *any* behavior encoder.
>
> 2. **Unlocking New Capabilities**: BUA's contribution goes far beyond improving accuracy. It has the ability to perform complex multi-tasks, including interpretable inference and behavior sequence generation (see Sections 3.4 and 4.3 for details). These are capabilities that traditional base models cannot achieve on their own.
>
> 3. **Standard Relevance and Future Adaptability:** The positive correlation between encoder quality and downstream performance is a widely accepted phenomenon in the field of modal alignment, as evidenced by related works such as **CoLLM [1]**. Our framework can correctly utilize the semantic richness of different encoders. Furthermore, this characteristic ensures the **future adaptability** of BUA: as more powerful behavioral models emerge, our framework can achieve higher performance through integration.
>
> **Conclusion:** Although a more powerful backbone network can lead to better absolute results, BUA consistently leverages the potential of the backbone network, demonstrating its effectiveness and flexibility as a general alignment framework.
>
> **References:**
>
> [1] Zhang, Yang, et al. "CoLLM: Integrating collaborative embeddings into large language models for recommendation." IEEE Transactions on Knowledge & Data Engineering (2025).
>
> ### **Response to Weakness-2: Granularity and Scope of Activity Modeling**
>
> We appreciate your comment. We address this by clarifying our research scope and future roadmap:
>
> 1.  **Focus on Semantic-Level Rhythms:** Our current research explicitly targets **high-level daily activities** (e.g., commuting, working) rather than micro-actions. This semantic granularity is essential for modeling long-term user intentions and lifestyle patterns, which are the core foundations for intelligent personal assistants and recommendation engines.
>
> 2.  **Future Expansion to Micro-Behaviors:** We acknowledge the value of nuanced behaviors. We plan to progressively extend the BUA framework to finer granularities in future work by collecting and processing multi-scale data, thereby bridging high-level intent with low-level execution.
>
> ### **Response to Question-1: Handling Missing or Incomplete Data**
>
> We thank you for your practical consideration. We address this concern by clarifying the nature of our data and our modeling objectives:
>
> 1.  **Inherent Completeness of High-Level Data:** Unlike raw sensor signals (e.g., GPS points) that suffer from frequent dropouts, our study focuses on **high-level daily activities** (e.g., *Dining*, *Working*). These discrete semantic events are inherently more robust and comprehensive. In our real-world datasets, these logs serve as reliable "anchor points" that define the user's primary timeline, making the framework less susceptible to minor noise or gaps found in fine-grained trajectories.
>
> 2.  **Focus on Pattern Understanding vs. User Profiling:** Our primary goal is to achieve robust behavioral prediction and reconstruction based on an understanding of general behavioral rhythms, rather than building highly personalized digital twin models for each specific user. Therefore, our current dataset is complete for learning to understand behavioral patterns and does not require perfect historical data for each user to train effectively.

---

> > ### Author Response · Authors · 2025-11-28
> > **Response-Part 2**
> >
> > ### **Response to Question-2: Handling Multi-User or Group Behavior**
> >
> > We appreciate your forward-looking question. While our current work establishes the foundation for individual behavior understanding, we have concretely planned to extend the BUA framework to capture **multi-user interactions and collective behaviors** in future work. We propose two specific approaches to achieve this:
> >
> > 1.  **Social-Context Instruction Tuning:**
> >     Utilizing the text interface of the BUA, we can explicitly integrate multi-user interaction signals into the **instruction context** via prompts. This allows LLM to capture the dynamics of user interactions without requiring structural changes to the model.
> >
> > 2.  **Interaction-Aware Modality Alignment:**
> >     We plan to introduce a dedicated **interaction encoder** (e.g., a graph neural network) in addition to the behavior encoder. By fusing interaction graph embeddings with individual sequence embeddings in the projection layer, this framework can directly align collaborative signals and group patterns with the inference space of the LLM.
> >
> > ### **Response to Question-3: Limitations of Freezing Model Components**
> >
> > We appreciate your inquiry. Our decision to freeze specific components is a strategic choice based on the balance between **training efficiency** and **performance**. We address the limitations and trade-offs as follows:
> >
> > 1.  **Standard Practice and Strategic Trade-off:**
> >     Freezing pre-trained components (such as the encoder) is a widely adopted paradigm in Multimodal Large Language Model (MLLM) research, as seen in foundational works like **LLaVA [1]**. It is a calculated trade-off: it significantly accelerates training and reduces memory overhead without substantially compromising performance. This strategy allows for efficient alignment in resource-constrained environments.
> >
> > 2.  **Proven Effectiveness:**
> >     Experimental results show that even with this freezing strategy, our method achieves state-of-the-art performance on all tasks. This indicates that the frozen encoder already provides sufficient semantic features for the LLM through the projection layer.
> >
> > 3.  **Potential for Further Gains:**
> >     We acknowledge that freezing imposes a constraint on the model's adaptability. If computational resources and training time were unlimited, **full-parameter fine-tuning** (unfreezing all components) could theoretically allow the encoder to better adapt to specific downstream tasks, potentially yielding further performance gains. We consider this an exciting direction for future exploration when more resources are available.
> >
> >
> > Furthermore, we have revised the paper(Section 3) to briefly explain that the strategy of freezing the training of certain components is a decision made as a trade-off between training effectiveness and performance.
> >
> > **Reference:**
> > [1] Liu, Haotian, et al. "Visual instruction tuning." Advances in neural information processing systems 36 (2023): 34892-34916.
> >
> >
> > Finally, thank you again for your valuable suggestions. We hope our response has addressed your concerns. If you have any further questions, we look forward to further communication.

---

### Author Response · Authors · 2025-12-03
**General Comment**

We sincerely thank all reviewers for their constructive comments. In this rebuttal, we have addressed all questions, conducted additional ablation studies, and revised the paper to improve clarity. Below is a summary of our contributions and key responses. For detailed discussions, please refer to our specific replies to each reviewer.

### 1. Overview and Recognition
To address the challenges of **long-tail behaviors, interpretability, and multi-task unification** in human behavior modeling, we propose **Behavior Understanding Alignment (BUA)**. This framework integrates LLMs with pretrained behavior models through a novel three-stage curriculum (Sequence Understanding $\to$ User Profiling $\to$ Self-Reflective Refinement). Furthermore, by employing a loss-balanced multi-turn dialogue setting, BUA establishes an implicit chain of thought—**"understanding before prediction|generation"**—which effectively enhances the model's predictive and generative capabilities. Experiments demonstrate that our method achieves state-of-the-art results on multiple real-world datasets.

We are encouraged that the reviewers recognized the effectiveness of our framework:
*   **Reviewer 9zNM** highlighted that BUA *"effectively bridges the modality gap... enabling unified handling of prediction and generation while improving long-tail performance."*
*   **Reviewer uFz2** noted the method *"tackles a timely challenge"* and praised the *"structured progression...contributes to interpretability."*
*   **Reviewer D24p** acknowledged that *"experimental results on two real-world datasets demonstrate the effectiveness of BUA."*

---

### 2. Methodological Clarifications & Theoretical Interpretations
We clarify key design choices and correct misconceptions regarding model architecture and mechanisms:

**A. Clarification on Input & Data Granularity**
*   **Focus on High-Level Activities:** We clarify that our behavioral data consists of semantically rich, high-level daily activities (e.g., "commuting," "dining") rather than raw, noisy sensor signals. This granularity is fundamental for effective semantic modeling.
*   **Hybrid Input Strategy:** BUA consistently utilizes a **hybrid input** (Text Instructions + Sequence Embeddings). The sequence encoder is always applied to **all data** to extract latent collaborative knowledge, correcting the misconception that it is used conditionally only for long sequences.

**B. Rationale for Freezing Components**
*   **Efficiency & Standard Practice:** Freezing the large behavior encoder is a strategic trade-off for training efficiency, which is a common technique in multimodal model training strategies (e.g., LLaVA-NeurIPS'2023).

**C. Qualitative Mechanism for Long-Tail Improvement**
*   **Semantic Rhythm vs. Frequency:** BUA improves prediction by understanding **semantic rhythm** rather than relying solely on occurrence frequency. For example, consider **"subway commuting."** While statistically sparse compared to habitual actions like "watching videos," it is semantically prominent as a transition signal (e.g., work $\to$ home). Through our *Key Behavior Identification* task, BUA explicitly captures this high-level pattern features. These features are then used in multi-turn dialogues to improve the prediction accuracy of such long-tail intentions..

---

### 3. Experimental Validations & Fairness
We provide new empirical evidence to validate our design choices and ensure fair comparisons:

**A. Necessity of Curriculum Learning (New Experiment)**
*   To address **Reviewer g36F**'s concern, we added a **Convergence Analysis** (Appendix S, Figure 4). The results demonstrate the necessity of our structured, easy-to-hard training progression, showing that the proposed **Staged Training** yields significantly better convergence on complex tasks compared to Joint Training.

**B. Fairness of Comparisons**
*   **Consistent Base Models:** All fine-tuned LLM baselines (TALLRec, CoLLM, LLaRa, etc.) utilize the **same backbone (Qwen2.5-7B)** and are trained on the **same dataset** as BUA. Additionally, comparisons with the non-fine-tuned base model confirm that performance gains are strictly attributable to our alignment framework.
*   **DeepSeek Benchmarking:** DeepSeek-V3 (671B) is included as a zero-shot SOTA reference to benchmark the gap between our specialized model and massive general LLMs, as fine-tuning such large models is computationally infeasible under standard settings. This setup is also a common practice in this field, such as Tallrec(RecSys'23).

**C. Clarification on Interpretability Metrics**
*   We clarify that we assess the model’s understanding and interpretability via **Qualitative Case Studies** and **Quantitative Human Evaluation** (Rationality & Interpretability scores), rather than using metrics like Accuracy or JSD for this purpose.

Finally, we believe these revisions and additional evidences firmly support the validity and novelty of BUA.

Best regards,
The Authors

---

### Note · Authors · 2026-01-06

I have read and agree with the venue's withdrawal policy on behalf of myself and my co-authors.